# Visualization of liquid-liquid phase transitions using a tiny G-quadruplex binding protein

Bikash R. Sahoo [1,2], Xiexiong Deng[1,2], Ee Lin Wong[1,2], Nathan Clark[1,2], Harry J. Yang [1,2], Alexey Kovalenko[1,2], Vivekanandan Subramanian[3], Bryan B. Guzman[4], Sarah E. Harris[5], Budheswar Dehury [6], Emi Miyashita[7], J. Damon Hoff [8], Vojč Kocman [9], Hirohide Saito [7,10], Daniel Dominguez [4], Janez Plavec [9] & James C. A. Bardwell [1,2] ✉

Liquid-liquid phase transitions govern a wide range of protein-protein and protein-RNA interactions. Although the importance of multivalency and protein disorder in driving these transitions is clear, there is limited knowledge concerning the structural basis of phase transitions or the conformational changes that accompany this process. In this work, we found that a small human protein, SERF2, is important for the formation of stress granules. We determined the solution NMR structure ensemble of SERF2. We show that SERF2 specifically interacts with non-canonical tetrahelical RNA structures called G-quadruplexes, structures linked to stress granule formation. The biophysical amenability of both SERF2 and RNA G4 quadruplexes have allowed us to characterize the multivalent protein-RNA interactions involved in liquid-liquid phase transitions, the role that protein disorder plays in these transitions, identify the specific contacts involved, and describe how these interactions impact the structural dynamics of the components enabling a detailed understanding of the structural transitions involved in early stages of ribonucleoprotein condensate formation.

When cells experience stressors that dissociate the translation complex, the resulting naked mRNA recruits proteins to form a liquid-liquid phase transition (LLPT) compartment known as the stress granule[1]. These membraneless organelles have been implicated in storing proteins and RNAs and protecting cells from stress-induced damage[2]. A key stress granule protein, G3BP1, recruits G-quadruplexes (G4s) to stress granules[3]. RNA G-quadruplexes (rG4s) are non-canonical RNA structures formed through Hoogsteen guanine (G-G) base pairs stabilized by a centrally located monovalent cation,

distinguishing them from canonical Watson-Crick base-paired structures[4]. These non-canonical structures have also been detected in the nucleus, cytoplasm, mitochondria, and the endoplasmic reticulum[5-7]. They have been linked to telomere maintenance, initiation of DNA replication, control of transcription and translation and protein folding[8-12]. Expansion of the rG4-forming G4C2 hexanucleotide RNA repeat sequences within the *C9orf72* gene has been linked to neurodegenerative diseases, including amyotrophic lateral sclerosis (ALS) and frontotemporal dementia (FTD)[13]. rG4s are thought to be

[1]Howard Hughes Medical Institute, University of Michigan, Ann Arbor, MI, USA. [2]Department of Molecular, Cellular and Developmental Biology, University of Michigan, Ann Arbor, MI, USA. [3]College of Pharmacy, University of Kentucky, Lexington, KY, USA. [4]Department of Pharmacology, University of North Carolina, Chapel Hill, NC, USA. [5]Department of Biochemistry and Biophysics, University of North Carolina, Chapel Hill, NC, USA. [6]Department of Bioinformatics, Manipal School of Life Sciences, Manipal Academy of Higher Education, Manipal, India. [7]Center for iPS Cell Research and Application, Kyoto University, Kyoto, Japan. [8]Department of Biophysics, University of Michigan, Ann Arbor, MI, USA. [9]Slovenian NMR Centre, National Institute of Chemistry, Ljubljana, Slovenia. [10]Institute for Quantitative Biosciences, The University of Tokyo, Tokyo, Japan. ✉e-mail: jbardwel@umich.edu

transiently folded in vivo[14]. However, in cells experiencing stress conditions, rG4s are enriched and bind to disordered proteins, including G3BP1[15–18]. Upon this binding, rG4s appear to be involved in the formation and organization of stress granules[13,17,18].

Despite the emerging importance of rG4s, relatively little is known regarding the regulation of their folding and function in vivo. Several proteins with disordered domains[19], including nucleolin[20], helicases[21,22], FUS[23], hnRNPA1[24], TDP43[25], TRF2[26], and Znf706[27] have been shown to interact with rG4s in vitro, to regulate their folding. Helicases, for example, are thought to unwind rG4s to safeguard against G4-induced genome instability[28], while nucleolin acts to enhance the stability of G4 structures[29]. Determining the mechanism and structural details of rG4-protein interactions will aid in understanding how proteins can regulate the functions of these noncanonical nucleic acid structures. However, our understanding of these interactions is complicated by the often-dynamic nature of the protein and rG4 components of these complexes and their tendency to undergo LLPT[19]. Recent studies have elucidated the structural mechanisms of a few proteins, such as helicases DHX36 and Pif1[30,31] and nucleolin[32], whose four RNA-binding domains bind DNA G4s to modulate their function. Despite these advances, the structural mechanisms by which proteins interact with rG4s to mediate ribonucleoprotein condensation via LLPT, or how the dynamic behavior of individual LLPT components is regulated, remain poorly understood. For example, though RNA molecules or intrinsically disordered proteins are identified as key components in LLPT, how proteins that can bind rG4s and mediate LLPT is unclear[33,34]. One way around this apparently intractable problem is to focus on NMR tractable components of stress granules, as NMR is unrivaled in its ability to provide detailed dynamic information[35]. However, NMR has so far only shown limited utility in analyzing LLPT[36]. Though >60% of proteins in the human proteome appear to contain disordered regions[37], most of these proteins are large, making them a challenge to analyze using NMR[36]. We decided to focus on small rG4s since several of them have detailed structural information available and since some appear to be involved in stress granule formation[15–17,38]. On the protein side, we chose to study the small NMR tractable protein SERF2[39], which we show here specifically binds to rG4s and is an important component of in vivo stress granules. Through studying SERF2 and rG4 interactions, we were able to obtain detailed insights into how SERF2 binds rG4 structures and potentially contributes to condensate formation.

## Results

### SERF2 binds to RNA G-quadruplexes

SERF family proteins were initially identified as in vivo drivers of amyloid formation[40–43], a process that has been linked to various age-related diseases[39–42]. However, their normal physiological functions remain obscure. SERF-related proteins share a conserved, highly charged N-terminal domain and are remarkably small, ~60–80 amino acids in length[43]. Previously, we showed that the SERF-related human Znf706 protein binds with high affinity to G4s primarily using N-terminal sequences that are conserved within the SERF protein family[27]. This motivated us to explore whether other SERF family proteins, such as human SERF2, also bind G4 structures. Using fluorescence polarization binding assays, we found that SERF2 binds with sub-micromolar affinities to various RNA sequences that are known to fold into rG4 structures[3,44], including polyG, (GGGA)4, (GGA)7, (G4C2)4 and (AGG)5. In contrast, SERF2 showed no interactions with guanine-rich RNA oligonucleotides that fail to form rG4 structures, as well as the homo-polyribonucleotide sequences polyA, polyC, or polyU, or single or double-stranded RNA sequences that form hairpin structures under physiological buffer conditions (pH 7.4, 150 mM KCl) (Fig. 1a, Supplementary Fig. 1). SERF2 shows ~10- to 20-fold stronger binding to rG4 forming repeat sequences such as four repeats of GGGGCC and human telomeric UUAGGG (TERRA23) as compared to their DNA G4 counterparts (Supplementary Fig. 2a, b). In contrast, an identical experimental setup demonstrated that SERF2 does not bind non-rG4-forming structures or unstructured RNAs, including HIV-1 TAR (5′-GGCAGAUCUGAGCCUGGGAGCUCUCUGCC-3′)[45], four-repeat CCCCGG[46] sequences and a mutated telomeric four-repeat sequence UUACCG[47] that is known not to form G4 structures (Supplementary Fig. 2c).

### RNA-binding sequence specificity of SERF2 in high-throughput sequencing

To further investigate the RNA-binding specificity of SERF-related proteins, two independent high-throughput binding assays were performed on SERF2, RNA bind-n-seq[48] and FOREST[49] (Folded RNA Element profiling with STructure). Both assays have been previously validated for known RNA-binding proteins, including those that specifically bind to rG4. The top five 6-mers obtained using the RNA bind-n-seq enrichment analysis are all very rich in guanines and are predicted to form stable rG4 structures[50] (Fig. 1c, d, Supplementary Fig. 2d). Using an independently done FOREST analysis[49], we found that 8 of the top 10 binding microRNA sequences were previously known to form rG4 structures (Fig. 1e).

The binding affinity of SERF2 for the FOREST hit (UG4U)6 and other previously studied rG4 sequences that are found in the cell, including some that are found in stress granules, was next determined using a fluorescence polarization assay. These rG4s include human telomeric TERRA repeat-containing RNA[51] TERRA23 (UAGGGUUAGGGUUAGGGUUAGGG), the four-repeat (GGGGCC)4 sequence (referred to as G4C2 hereafter), a hexanucleotide repeat found within the *C9orf72* gene, whose expansion is associated with ALS[52], and the mRNA G4s encoded by genes *Mark2* (5′-GAAGGGGAGGGGGCUGGGGGGGGGCAGGG-3′) and *Stxbp5* (5′-GGGAAGGGAAGGGGAGUGGG-3′) both of which are known to localize to stress granules[53,54]. SERF2 binds to all of these sequences with low micromolar affinity with $K_D$ values for rG4s G4C2 was $0.88 \pm 0.1\,\mu M$, for TERRA23: $0.30 \pm 0.02\,\mu M$, for UG4U: $0.73 \pm 0.25\,\mu M$, and for *Mark2*: $4.5 \pm 0.50\,\mu M$ (Fig. 1f, Supplementary Fig. 2e, f). These binding affinities are similar to the affinities of other rG4 binding proteins for their rG4 binding partner, including cold-inducible RNA-binding protein[55], FUS[56], and FMRP[57], as well as G4 binding small molecules such as pyridostatin, NMM, and BRACO-19[58]. We conclude that SERF2 specifically binds to rG4 structures. This is consistent with our earlier findings that the SERF-related protein Znf706 specifically binds to G4 sequences[27].

### SERF2 forms liquid-liquid phase transition droplets with RNA G-quadruplexes

In isolation, SERF2 remains soluble even at concentrations as high as 1 mM, in various salt concentrations (0–200 mM KCl) and under crowding conditions (10 % w/v PEG8000) (Supplementary Fig. 3a). Under non-crowding conditions, SERF2 also shows no phase separation when mixed with total HeLa cell RNA or rG4s (Fig. 2a–c, Supplementary Fig. 3b). However, in the presence of crowding agent PEG8000, we found that SERF2, even when present in concentrations as low as 1.5 to 3 μM, form irregularly shaped particles when mixed with total HeLa cell RNA (Fig. 2a). To understand the physical properties of these particles, fluorescence recovery after photobleaching (FRAP) measurements were carried out. FRAP results show that particles containing SERF2 interacting with total RNA are reversible and dynamic in nature with a recovery half-time of ~13 s (Fig. 2b). This observation indicates that the irregularly shaped SERF2-total RNA particles are similar to mesh-like condensates that are observed in some RNA-binding proteins[59,60]. In contrast, SERF2 forms spherical phase-separated droplets in the presence of different rG4-forming sequences in crowding conditions that include TERRA23, (G4C2)4, and (UG4U)6 (Fig. 2c). However, these rG4 sequences, when tested in isolation, do not phase separate on their own (Supplementary Fig. 3b,) suggesting SERF2 and rG4 interaction mediates phase separation.

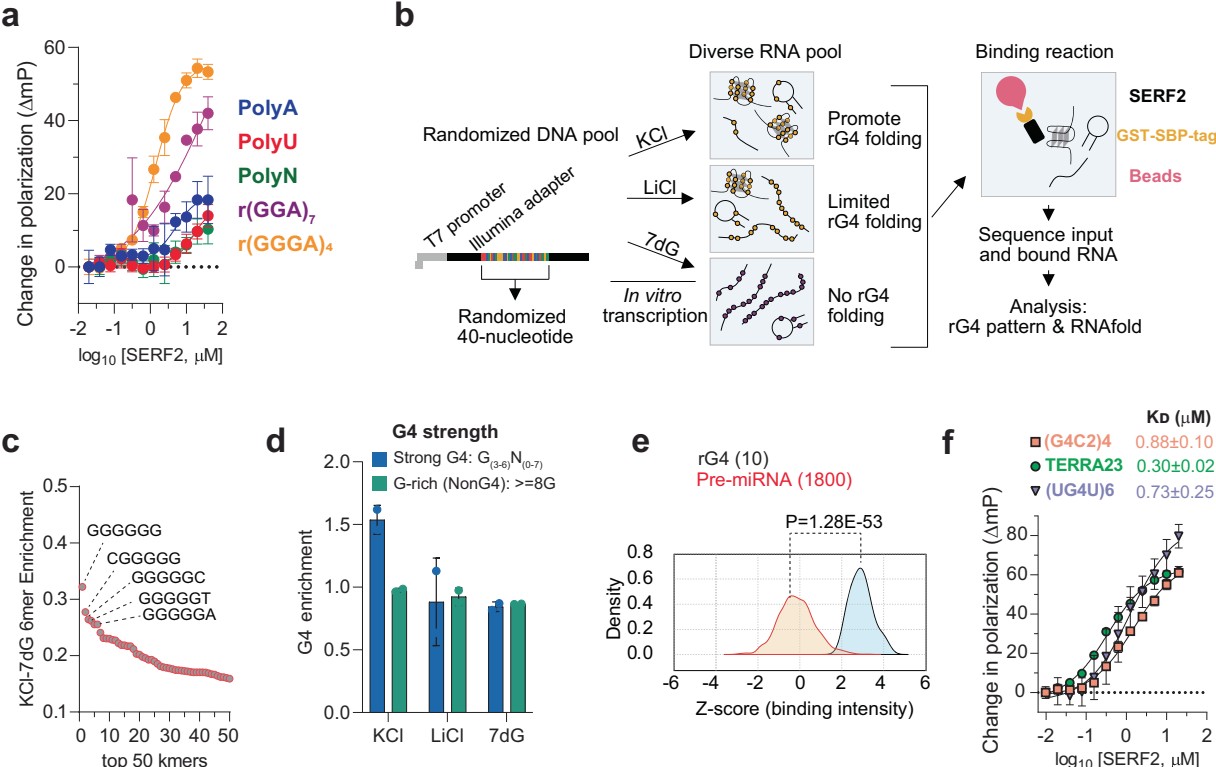

**Fig. 1 | High-throughput screening for SERF2 binding specificity. a** Fluorescence polarization assay used to measure the binding affinity of SERF2 to 6-FAM labeled poly ribo-polynucleotides and rG4 quadruplex-forming sequences, as indicated. Data are presented as mean values ± standard deviations obtained from three independent replicates. **b** A schematic representation of RNA Bind-n-Seq experiments using RNA pools. A randomized DNA oligonucleotide pool was transcribed to RNA and folded in a buffer containing either KCl, which favors rG4 formation, or LiCl, which disfavors rG4 folding. An additional RNA pool was made by replacing guanines (G) with 7-deaza (7dG) to eliminate rG4 quadruplex folding. These pools were mixed with GST-SBP-SERF2, and bound RNA was isolated and sequenced with ~2 × 10⁶ reads. **c** RNA Bind-n-Seq analysis of the 6-mers enrichment in KCl versus RNA that was made with 7dG in a sample mixture containing 50 nM SERF2. The guanine-rich 6mers in the top 5 kmers are labeled. **d** Enrichment of rG4 quadruplex patterns in different conditions was classified by G4 quadruplex strength in buffer containing 50 nM SERF2. Sequences containing 3 or more guanines in the G-tetrad are referred to as strong rG4s, while sequences with ≥8 guanines but lacking a defined G4-forming motif as defined by QGRS mapper program are referred to as non rG4s. Data are presented as mean values ± standard deviations for each condition obtained from two different RNA library preparations. **e** FOREST analysis: average binding intensities of SERF2 to a library containing 1800 folded human pre-miRNAs (orange curve) and to 10 defined rG4 structures (gray curve). The *p*-value shown was determined by the two-tailed Brunner-Munzel test. **f** The fluorescence polarization plot that shows binding affinity measurements between SERF2 and three different 6-FAM labeled rG4s. The binding assays in (**a** and **f**) were done with varied protein concentrations mixed with 20 nM rG4 quadruplex or polynucleotides at room temperature in 20 mM NaPi (pH 7.4), 100 mM KCl. The standard deviations shown were calculated from three independent replicates, and data are presented as mean values.

Phase regime studies revealed that SERF2-rG4 droplets form at a variety of concentrations of salt, PEG8000, protein, and TERRA23 rG4 (Fig. 2d, Supplementary Fig. 3c). The 1.5–3 μM concentrations of SERF2 that allow phase separation in the presence of rG4s are below the SERF2 concentrations present in human cells which range from ~7 μM in U2OS cells to ~23 μM in MCF7 breast cancer cells[61]. SERF2 also phase separate with a small 12-nucleotide-long TERRA12 rG4 but does not phase separate when mixed with the similarly sized non-rG4-forming 10- or 20-nucleotide polyA RNA (Supplementary Fig. 3d). We postulate that rG4 structures likely trigger SERF2 binding that drives LLPT.

## SERF2 and RNA G-quadruplexes form slowly exchanging droplets

rG4s have recently been shown to regulate LLPT, and in neurodegenerative diseases, there is evidence that they accumulate[62] and may act as molecular scaffolds that help transition liquid droplets to more solid gel-like aggregates[63]. We previously showed that G4s promote α-synuclein aggregation, a process that is associated with Parkinson's disease[27], but suppress the ability of the SERF-related protein Znf706 to promote α-synuclein aggregation by promoting Znf706's ability to form LLPT. Recently, TERRA rG4s have also been shown to promote α-synuclein aggregation in vivo[47], indicating that G4s regulate protein

aggregation. Our observation that SERF2's interaction with rG4 can lead to LLPT is intriguing and suggestive of a link to the possible roles of SERF2 and rG4 in neurodegenerative diseases. To explore how rG4 binding may affect SERF2's transition to the liquid-liquid phase, we examined the dynamics of the interactions between SERF2 and RNA or rG4 using FRAP experiment. A recovery with a half-time of ~30 s was observed for SERF2 condensates formed with short (30 nucleotide) or long (700–3500 kDa) random length polyA (Supplementary Fig. 3e, f). In contrast, SERF2 droplets in the presence of structured RNA that include rG4s or hairpin HIV-1 TAR RNA that bind weakly to SERF2 in solution were found to recover more slowly (Supplementary Fig. 3h). In the presence of rG4s, SERF2 recovery half-time was ~ 49, 70 and 79 s for TERRA23, (G4C2)4 and (UG4U)6 rG4s respectively, which is similar to recovery times observed for the non-rG4 hairpin structure of HIV-1 TAR RNA (~ 49 s) (Fig. 2e, Supplementary Fig. 3g, h). These results suggest that SERF2 droplets are less fluid when formed with structured RNA binding as compared to SERF2 droplets that form with unstructured RNAs.

## Structural characterization of SERF2 reveals partial disorder

Studying the detailed structure of ribonucleoprotein condensates is challenging due to their dynamic and heterogeneous nature[64], and

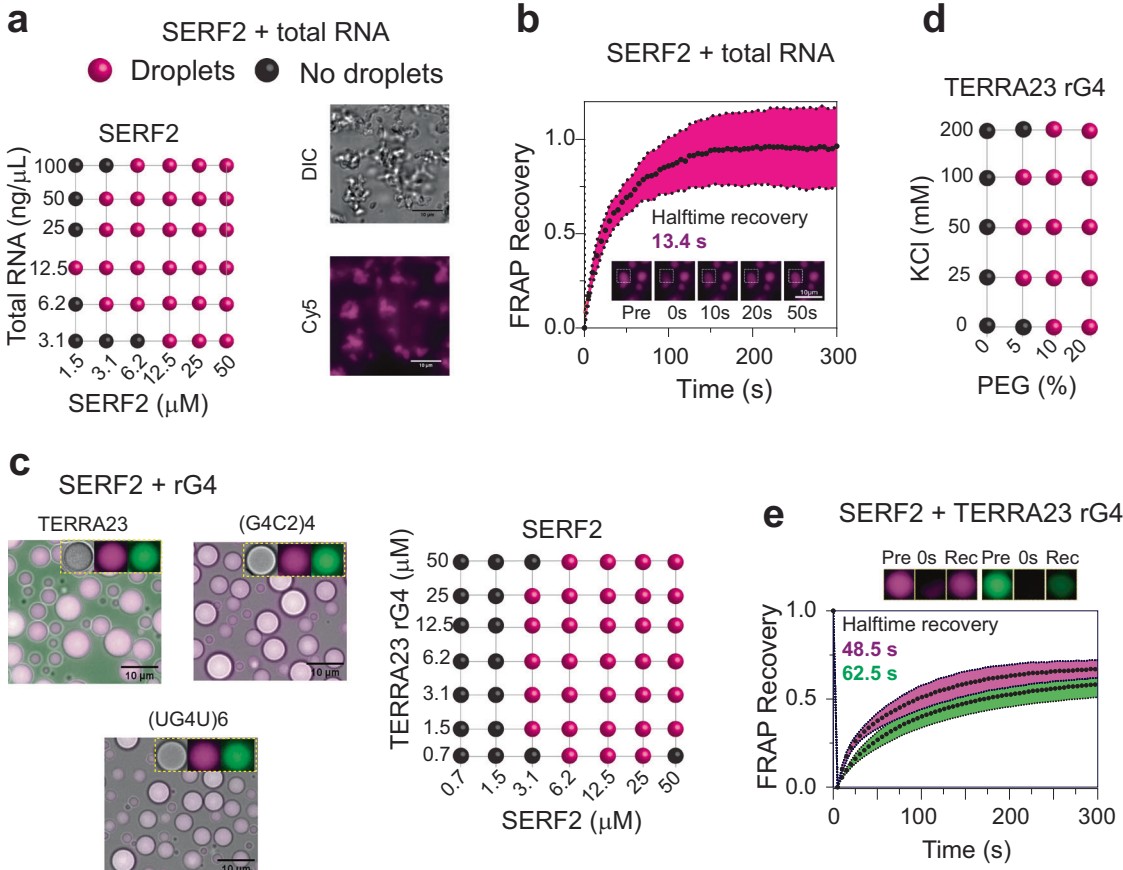

**Fig. 2 | RNA interaction drives liquid-liquid phase transition in SERF2. a** Phase regime illustrating phase transitions for SERF2 as a function of total RNA concentration using RNA extracted from HeLa cells (left panel). Fluorescence imaging shows gel-like structures present in solutions containing 50 μM SERF2 mixed with 200 ng of total HeLa cell RNA containing 10% (w/v) PEG8000, incubated for 30 min at room temperature (right panel). **b** Dynamics and recovery of Cy5-labeled SERF2 in total HeLa cell RNA droplets, obtained by FRAP analysis, suggest that the mesh-like condensates observed in (**a**), right panel, are dynamic and reversible. Standard deviations were calculated by analyzing 6 isolated droplets subjected to FRAP, and data are presented as mean values. **c** Fluorescence images show 50 μM SERF2, dissolved in 20 mM NaPi (pH 7.4), 100 mM KCl, readily undergoes phase transition (right) when mixed with equimolar concentrations of three different rG4s, TERRA23, (G4C2)4, and (UG4U)6. Similar results were obtained in three

independent repeated experiments. The sample mixture contains 0.5% Cy5 labeled SERF2 (purple) and 6-FAM rG4 (green), as indicated in the figure inset. Phase regimes illustrating phase transitions for SERF2 and TERRA23 rG4 in 10% w/v PEG8000, at varying protein and RNA concentrations, are shown on the right panel. **d** Phase diagram showing SERF2 and TERRA23 rG4 sample mixtures undergoing phase separation, in varying salt and PEG8000 concentrations. **e** Two-component FRAP analysis was done to measure the recovery rates of SERF2 (purple) and TERRA23 rG4 (green) in SERF2-rG4 droplets. Data are presented as mean values, and standard errors were calculated by analyzing 8 isolated droplets subjected to FRAP. The pre-bleached, after-bleached (0 s), and recovered droplets (300 s) are shown above the FRAP plot. The FRAP data were fitted in GraphPad Prism, using a non-linear regression, one-phase association model, to obtain the recovery half-time (t1/2) reported in the text.

often the large size and complex multi-component composition of these condensates. One challenge is the structural flexibility of the components involved, many of which contain regions of intrinsic disorder[65]. Though reviews have highlighted the importance of intrinsically disordered proteins in many biological systems[66,67], there is an understandable paucity of experimental approaches to examine how intrinsically disordered proteins interact with other cellular components. To address this gap, we focused on the interaction between the partially disordered protein SERF2 and rG4 structures. This system provides a tractable model for structural studies, as both SERF2 and rG4 species, such as the G4C2 repeat, and the telomeric-repeat TERRA10 or TERRA12, are NMR-compliant, and NMR is well-suited to study flexible interactions[68].

While TERRA12 and TERRA10 rG4 structures are very well-characterized[69–71], the structural properties of SERF2 are relatively poorly characterized. Thus, to study SERF2-rG4 interactions, including those involved in LLPT, we first explored the structural properties of SERF2. SERF2 presented a narrow 2D 1H-15N HSQC NMR chemical shift dispersion at 4 °C, indicating that it contains a high degree of

structural disorder even at this low temperature (Supplementary Fig. 4a). At 37 °C most of the 1H-15N peaks disappear from the HSQC spectrum, suggesting that a higher degree of disorder exists for SERF2 at this temperature or that induction of different conformational states in SERF2 occurs upon temperature upshift, a phenomenon that has been demonstrated to occur in other disordered proteins[72,73] (Supplementary Fig. 4a). Circular dichroism spectroscopy analysis indicates that substantial helicity exists in the SERF2 structure at 4 °C, with decreasing helicity as the temperature increases (Supplementary Fig. 4b). The well-resolved peaks observed for SERF2 in multi-dimensional NMR experiments performed at low temperature (4 °C) and at low pH (5.5) or both, enabled us to achieve NMR backbone assignment for 51 of the 58 non-proline residues[39] found in SERF2 and to perform torsion-angle analysis on SERF2 using TALOS-N[74]. Using a total of 558 restraints that include 40 dihedral angles and 24 hydrogen bonds (Table 1) and 494 NOE restraints (intra−205, short−220 and medium−69), we calculated an ensemble of structures for SERF2 (Fig. 3a). The final ensemble shows good structural quality in the ordered region (residues 32–48), with a backbone RMSD of

**Table 1 | NMR structural statistics of human SERF2**

| Parameters | Apo-form |
|---|---|
| **Restraints** | |
| NOE (Total) | 494 |
| Intra-residue (\|i-j\| = 0) | 205 |
| Short (\|i-j\| < 1) | 220 |
| Medium range (2 < \|i-j\| < 5) | 69 |
| Long range (\|i-j\| > 5) | - |
| Hydrogen bond | 24 |
| Dihedral angle | 40 |
| Total restraints | 558 |
| Restraints per residue (total/long range) | 9.5 |
| **Residual distance restraint violations** | |
| Target function | 0.45 ± 0.17 |
| Average distance violations/structure (Å) | 0.02 ± 0.01 |
| Average angle violation/structure (°) | 0.25 ± 0.08 |
| Maximum distance violation (Å) | 0.67 ± 0.12 |
| Model quality (ordered region 32–48) | |
| RMSD from average coordinates (Å) | |
| All backbone atoms (ordered/all) | 0.24 ± 0.09 |
| All heavy atoms (ordered/all) | 1.15 ± 0.20 |
| Molprobity Ramachandran plot | |
| Most favored regions (%) | 75.2 |
| Additionally allowed regions (%) | 24.3 |
| Generously allowed regions (%) | 0.5 |
| Disallowed regions (%) | 0.0 |

$0.24 \pm 0.09$ Å, and no distance violations >0.5 Å or Ramachandran outliers. We found that the N-terminal (1–32) and C-terminal (48–59) regions in SERF2 to be dynamic and disordered, while residues 33–47 adopt an ensemble-averaged helical conformation (Fig. 3a, Supplementary Fig. 4c). The structural ensemble of the helical core and disordered flanks observed in SERF2 is shared with its sequence homolog, human Znf706 which also has a disordered N-terminal domain, though Znf706 contains a C-terminal zinc finger that SERF2 lacks[27].

## Disorder in SERF2 mediates its interaction with RNA G-quadruplexes

Disordered regions are commonly involved in protein-protein and protein-RNA interactions[75], but the exact role that disorder plays in these interactions is often unclear. Studying the molecular interactions between TERRA rG4 and SERF2 provides a well-suited model to elucidate the mechanistic contributions of disorder to RNA recognition and condensate formation. TERRA rG4 was selected because it has been previously structurally well-characterized[70,76] and because of the evidence for its presence in cellular condensates like stress granules, and its potential involvement in their formation and organization[53]. We employed NMR to explore in detail the interaction between SERF2 and TERRA rG4, the role played by SERF2's disordered domain and the structural impact of binding in both interacting partners. The long repeat TERRA rG4s, such as TERRA23 or TERRA60, bind stronger to SERF2 as compared to short TERRA12 or TERRA10 repeats (Fig. 1f, Supplementary Fig. 5a). However, the short TERRA12 or TERRA10 rG4 sequences fold better and form multimeric structures suitable for structural studies, as demonstrated earlier[69]. TERRA12 in 20 mM NaPi, 100 mM KCl, pH 7.4 buffer that presented a strong CD absorption at ~263 nm and well-resolved imino proton NMR peaks (Supplementary Fig. 5b), indicating that it is well folded compared to TERRA23[64]. Studying different TERRA rG4 sequence lengths enabled us to investigate how sequence influences rG4 structure and RNA-protein

interactions, with some lengths being more useful to address certain specific questions. For instance, we used both longer and shorter TERRA rG4 sequences to map RNA-binding sites in SERF2, but primarily used shorter sequences such as TERRA10 and TERRA12 to map protein-binding sites on RNA due to their better NMR spectral resolution. In contrast, longer TERRA sequences such as TERRA23 and TERRA60, though structurally more heterogeneous and challenging to characterize in detail, are biologically relevant given their intramolecular G4 structure[26], association with telomeric chromatin, involvement in scaffolding amyloid aggregation and evidence suggesting their role in stress granule formation and genome regulation[47,53,77]. The TERRA10, TERRA12 and TERRA23 sequences bound to a similar region of SERF2 at moderate to low RNA to protein ratios (Fig. 3c, Supplementary Fig. 5c, d). The longest TERRA rG4, TERRA60, showed signal broadening in several SERF2 residues (Supplementary Figs. 5d, 6) at very low concentration (1:40 RNA:protein) that increases upon increasing RNA concentration (1:4 RNA:protein). This observation is similar to what was observed for equimolar or high RNA concentrations for short and long TERRA rG4 sequences such as TERRA23, TERRA10 and (Supplementary Fig. 6), consistent with the hypothesis that various length TERRA rG4 lengths bind to SERF2, but with affinities that increase with TERRA length.

The residue-specific binding site analysis on SERF2 by TERRA rG4 was next probed using TERRA23 a length that binds relatively stronger as compared to TERRA12, and shows a sub-micromolar binding affinity similar to TERRA60. Further, at low RNA to protein ratio, it yields small NMR signal broadening, enabling 1H-15N HSQC titration experiments. TERRA23 rG4-SERF2 interaction causes substantial chemical shift perturbations both in SERF2's N- (3–21) and C-terminal (51–56) residues, as tracked by 1H-15N HSQC spectra (Fig. 3b). At low TERRA23 rG4 concentrations (Fig. 3c, green), SERF2 residues R7, R11, K16, K23, R27, Q46, K47 and K54 showed substantial chemical shift changes suggestive of direct binding to rG4 or conformational alteration upon rG4 interaction. The N-terminal domain residues of SERF2, including T2, N5, R7, R11 and S21, showed increasing perturbation as the rG4 concentration increased (Fig. 3b), implicating the N-terminal domain of SERF2 of involvement in rG4 binding. Binding assays showed that the isolated N-terminal domain of SERF2 (residues 1–32) binds TERRA23 rG4, though ~10-fold weaker than the full-length protein. Conversely, the C-terminal domain of SERF2, residues 31–59, in isolation, showed no binding saturation upon addition to TERRA23 rG4 quadruplex in a fluorescence polarization binding assay (Supplementary Fig. 7a). These results suggest the N-terminal domain of SERF2 drives rG4 recognition, but both domains coordinate for high-affinity binding, forming a tight SERF2-TERRA rG4 complex.

To further explore the impact of TERRA rG4 binding on SERF2 dynamics at residue level resolution, we compared the NMR relaxation data of SERF2 in its free and rG4-bound states. The relaxation measurement results showed that TERRA23 rG4 binding significantly increases $R_2$ values in both the N-terminal and C-terminal intrinsically disordered regions of SERF2, suggesting that SERF2 undergoes segmental slow-motion (Supplementary Fig. 7b). In contrast, $R_1$ and heteronuclear NOE values remain largely unchanged, suggesting that SERF2 maintains local flexibility after interaction with TERRA23 (Supplementary Fig. 7c, d). The dynamic and disordered SERF2 regions spanning residues 3–24 and 48–56, identified by their low heteronuclear NOE values (<0.3), showed significant chemical shift perturbations and reduced signal intensities upon TERRA23 rG4 interaction (Supplementary Fig. 7d, e). These dynamics suggest that flexibility in these regions facilitates rG4 recognition. Binding also led to increased spin-spin relaxation rates (T2) and higher backbone R2/R1 values, indicating restricted SERF2 dynamics in the SERF2-TERRA23 rG4 complex compared to the unbound state (Fig. 3d). The α-helical core of SERF2 (residues 37–47) showed a marked increase in $R_2$ values but no significant changes in $R_1$ or heteronuclear NOE values, indicating local conformational exchange or slow dynamics

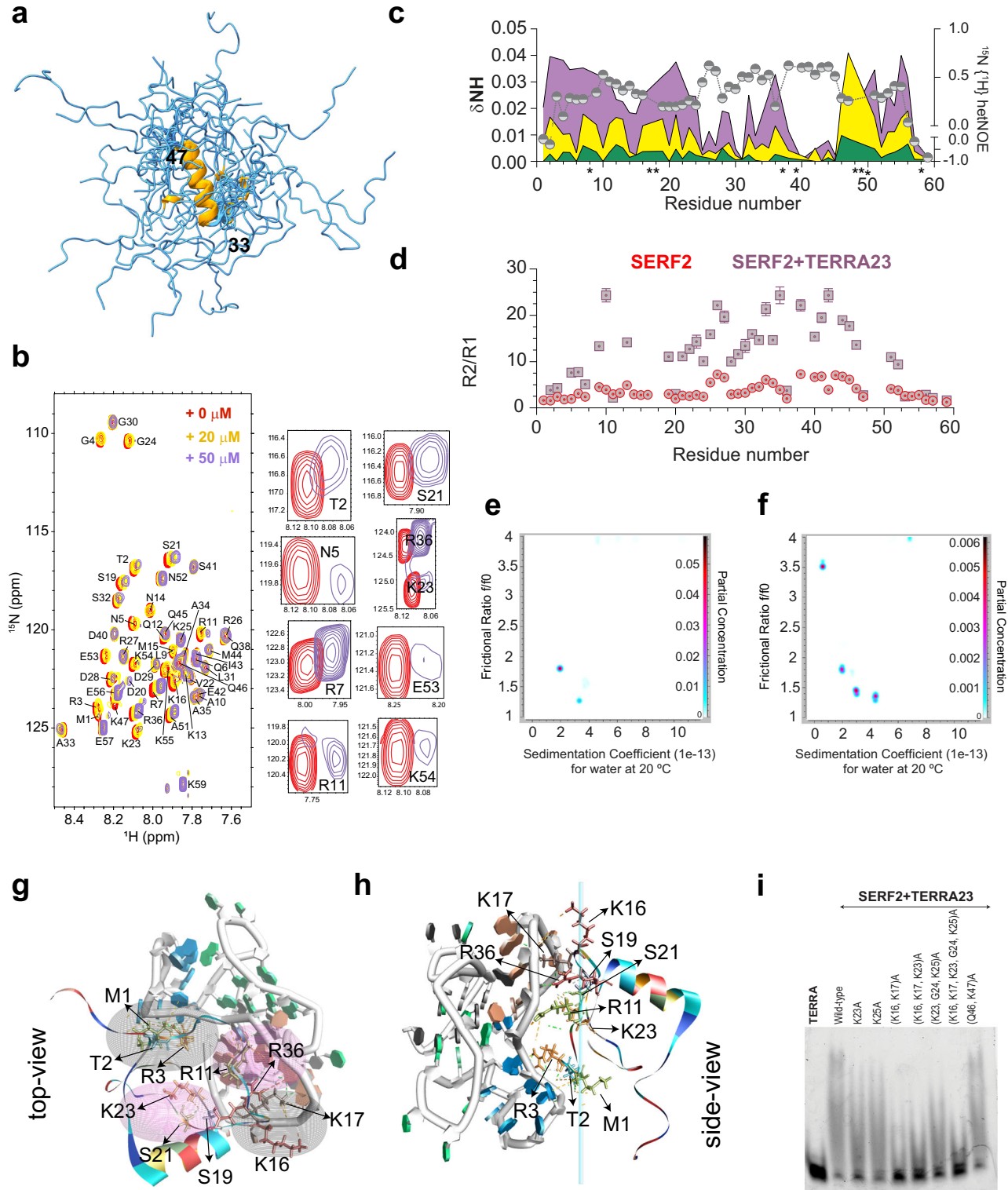

in the intermediate regime occur upon RNA binding, rather than changes in backbone order or in secondary structure. Protein secondary structure analysis using CD spectroscopy confirmed this, no substantial change in SERF2 protein helicity was present in the presence of TERRA23 rG4 (Supplementary Fig. 7f). The slowed dynamics and peak broadening in SERF2 upon TERRA23 rG4 binding (Supplementary Fig. 6) could arise from an increase in size that occurs upon protein-TERRA23 rG4 association to form either dimers or higher oligomeric forms. Size-exclusion chromatography analysis that showed that mixing SERF2 with TERRA23 rG4 generates multimeric species with distinct populations that are not

present in solutions containing SERF2 or TERRA23 in isolation. A 1:1 SERF2-rG4 complex of ~30 kDa, and higher-order multimeric complexes that are >30 kDa (Supplementary Fig. 7g). These complexes are detected by rG4 absorption at 260 nm, where SERF2 has no absorption.

We next tested if SERF2 has preferential binding sites within the TERRA rG4 using the structurally well-characterized TERRA10 and TERRA12 rG4 sequences. These lengths were tested because they exhibit well-resolved guanine imino protons in comparison to TERRA23's poorly resolved spectra. Binding of SERF2 caused chemical shift changes in the imino protons of G4, G5, G9, and G10 of TERRA12 and

**Fig. 3 | High-resolution NMR structure of SERF2 reveals that its disordered and dynamic N-terminal domain binds TERRA rG4. a** 20 best NMR ensemble model structures of SERF2. The average converged helical structure spanning residues 33–47 in SERF2 is shown as the helix in orange. **b** 1H-15N NMR assignment (red spectrum) of 100 μM human SERF2 mixed with 20 μM (yellow) and 50 μM (purple) TERRA23 rG4 at 4 °C. Spectral zooms on the right illustrate chemical shift changes in several N- and C-terminal residues at 2:1 protein: rG4 ratio. **c** The 1H-15N chemical shift perturbations (left y-axis) were calculated from (**b**) and plotted as color-shaded peaks for each assigned residue in SERF2 at increasing TERRA23 rG4 concentrations. The yellow and purple colors in the graph correlate to their corresponding spectrum shown in (**b**); unassigned peaks are denoted with an asterisk. The heteroNOE values of SERF2 in the absence of TERRA23 rG4 were plotted to highlight the dynamic regions in SERF2. **d** 15N relaxation rates R2/R1 demonstrating that a significant change in dynamics occurs for 200 μM SERF2 on its own (pink

dots) and in the presence of 100 μM interacting TERRA23 rG4 (gray squares), as a function of residue number. Error bars represent the standard error from per-residue exponential fitting of peak intensities in NMRFAM-Sparky. **e, f** 2D analysis plots derived from analytical ultracentrifugation experiments for 4.7 μM TERRA12 rG4 without (**e**) or mixed with a 2-molar excess SERF2 (**f**). The partial concentration shown in color on the right y-axis represents the abundance of individual species in the sample solution. **g, h** Cartoon shows the top (**g**) and side-view (**h**) of SERF2 and TERRA rG4 complex. Note the quadrupole-like and planar interactions shown respectively as ellipses in (**g**) and vertical slabs in (**h**). Residues generating the quadrupole-like interactions and distorting TERRA rG4 structure (PDB ID: 2M18) are labeled, and hydrogen bonds are indicated with dashed lines. **i** EMSA gel-shift assay of 5 μM TERRA23 rG4 mixed with an equimolar amount of wild-type SERF2 and different lysine to alanine SERF2 mutants. Similar results were obtained in two independent repeated experiments.

TERRA10 (Supplementary Fig. 8). This observation indicates SERF2 may either directly bind near TERRA10 or TERRA12 rG4's G-quartet core, or alternatively that SERF2 binding may impact G-quartet conformation (Supplementary Fig. 8). To further probe the spatial proximity between SERF2 and the TERRA12 rG4, saturation transfer difference (STD) NMR experiments were carried out. STD NMR approach detects interactions between protein and rG4 by transferring selective radiofrequency saturation from one binding partner to the other, indicating close spatial proximity. When guanine imino proton (-11.11 ppm) of TERRA12 rG4 was selectively saturated, magnetization transfer was observed across both amino (-7–8.5 ppm) and aliphatic (-0.5–3 ppm) regions of SERF2-rG4 complex (Supplementary Fig. 9a). These transfers were absent when TERRA12 rG4 was examined in isolation by saturating different imino peaks, indicating that the observed saturation arises from complex formation (Supplementary Fig. 9b). When the amino proton of SERF2, particularly the signal corresponding to residue A33 at -8.5 ppm (isolated from RNA amino peaks, Fig. 3b), was selectively saturated, transfer to the RNA imino region was also observed (Supplementary Fig. 9c). The STD effects were observed in the -11–11.5 ppm region, corresponding to the guanine imino protons of TERRA12 rG4, indicating that magnetization was transferred from SERF2 to these protons upon complex formation. Importantly, these effects were absent in control experiments using either SERF2 or TERRA12 alone (Supplementary Fig. 9b, d). Together, these results support a model in which SERF2 binds in close spatial proximity to the G-quartet core of TERRA12 rG4, enabling intermolecular magnetization transfer being detectable by STD NMR.

## SERF2 and RNA G-quadruplexes form multimeric complexes

Since SERF2 forms multimeric protein-RNA assemblies and is involved in RNA-mediated phase transitions, we next investigated the mechanisms of SERF2-RNA complex formation and how multimerization occurs. To determine the stoichiometry of these complexes, we used analytical ultracentrifugation, which provides higher mass resolution as compared to size-exclusion chromatography. Analytical ultracentrifugation of TERRA12 rG4 in isolation shows two structures with sedimentation coefficients of 1.95 and 3.33, suggestive of globular, folded structures (Fig. 3e). In the presence of SERF2, sedimentation coefficient values were observed with predicted sizes of -12.4, 17.4, and 27.1 kDa corresponding to monomeric rG4, 1:1, and 1:2 rG4-SERF2 complexes (Fig. 3f). These findings demonstrate that SERF2 forms multimeric complexes with TERRA12 rG4, supporting a model in which SERF2-rG4 interactions drive multimerization. Notably, similar multimerization behavior was observed when SERF2 was mixed with the longer TERRA23 rG4 (Supplementary Fig. 7g).

## Structural dynamics and stability of the SERF2-RNA G-quadruplex complex

The TERRA rG4 quadruplex binding sites in SERF2 we obtained from NMR measurements were used to build the TERRA-SERF2 complex

structure and subjected to an all-atom molecular dynamic (MD) simulation study. The goal of this analysis was to characterize the structural changes and dynamic interactions that occur during the early stages of SERF2-rG4 complex formation and their potential involvement in liquid-liquid phase transitions. To model the complex, the TERRA rG4 sequence length was selected based on available high-resolution structures, TERRA12, which forms a dimeric quadruplex (PDB 2KBP), and TERRA10, which adopts a tetrameric rG4 conformation (PDB 2M18)[78]. NMR experiments revealed that both TERRA10 and TERRA12 rG4s bind to a similar surface on SERF2, involving conserved guanine residues such as G4, G5, G9, and G10. Despite differences in higher-order folding with TERRA12 forming a dimeric quadruplex and TERRA10 adopting a tetrameric conformation, the local structural elements involved in SERF2 binding appear to be preserved. This observation supports the use of either RNA construct to model protein-RNA interaction interfaces. Given that in 100 ns all-atom MD simulation, TERRA10 adopts a more compact and stable conformation as compared to TERRA12 a rG4 that showed a relatively distorted end (Supplementary Fig. 10a). We therefore chose to use TERRA10 for the long-range MD simulation studies. Notably, MD simulation is well-suited to probe the conformational flexibility and binding dynamics of protein-RNA complexes, and remains informative regardless of differences in global topology. We consider that the choice of TERRA10 does not compromise the broader relevance of the model. Using the HADDOCK program[79], models of the TERRA rG4 and SERF2 1:1 and 1:2 complexes were built. We identified residues in SERF2 that are involved in direct or indirect binding (due to conformational alteration upon rG4 interaction) as evidenced from NMR chemical shift perturbations performed under non-crowding conditions. We also identified rG4 guanines that interact with SERF2 and show magnetization transfer in NMR saturation transfer difference experiments (Fig. 3b and Supplementary Fig. 9). These interacting protein residues and guanines were used as input to HADDOCK to generate SERF2-rG4 complex structures (see "Methods"). These models were then subjected to microsecond-long all-atom MD simulations to further examine the stability and dynamic nature of the complex structures. After a 0.5 μs all-atom MD simulation, the 1:1 rG4:SERF2 complex retained N-terminal contacts but lacked C-terminal interactions (Fig. 3g). In contrast, the 1:2 rG4:SERF2 complex gained additional C-terminal contacts (Supplementary Fig. 10b), aligning with our NMR data that showed chemical shift perturbations for several C-terminal residues upon rG4 binding. These observations illustrate specific multivalent interactions that occur between SERF2 and rG4 during an oligomerization process that can lead to LLPT.

Our analysis revealed that a hydrophilic core in the SERF2-rG4 complex is formed by residues in regions that are dynamic in monomeric SERF2 (Fig. 3g, h). Specifically, N-terminal residues in SERF2, including R3, R11, K16, K17, and K23, form hydrogen bonds with TERRA10 rG4 nucleotides G1, G3, G7-G9 that are located in the bottom two G-quartets. This interaction forms a planar interface in the SERF2-

TERRA10 rG4 complex. It consists of SERF2 residues M1-T2 and R3 facing opposite to K16-K17 in one interface, and residues R11-R36 facing residues S19-S21- K23 in another interface, creating a quadrupole-like contact architecture with alternating positive and negative charge residues distributed on the corners of a square (Fig. 3h). The guanines in the TERRA10 rG4 structure (PDB 2M18) form five stacked G-quartets through an 'i' to 'i + 6' nucleotide pattern[76] (Supplementary Fig. 10c). SERF2 binding in the 1:2 rG4:SERF2 complex disrupts two G-quartets, strongly distorting the rG4 structure. Residues R11 and R3 break the rG4 G9-G3 Hoogsteen bond, while S19-S21-K23 and M1-T2 interactions expose the rG4 U4, which stabilizes the structure via rG4 G2-G3 hydrogen bonding (Fig. 3h, Supplementary Fig. 10b). Additionally, K16-K17 bonding with TERRA10 rG4's G7 and G8 nucleotides disrupts G7-G1 and G7-G8 Hoogsteen pairs, further distorting the TERRA10 rG4 structure. Prior to binding SERF2, TERRA10 rG4 forms a parallel topology, and after binding, TERRA10 rG4 structure is noticeably distorted in the MD simulation.

The hydrophilic core of the 1:2 rG4:SERF2 complex is larger than that present in the 1:1 complex due to charged C-terminal residues (R36, K50, K54, K55) that form hydrogen bonds with TERRA10 rG4. Over the atomic simulation timescale, SERF2 shifts from interacting with uracil in the 1:1 complex to guanine in the 1:2 complex (Supplementary Fig. 10d). rG4 structure distortion was also observed in the 1:2 rG4:SERF2 complex. Experimental evidence backing up these MD simulation results includes a decrease in TERRA10 rG4 parallel structure with increasing SERF2 concentrations as measured by CD spectroscopy, and a co-commitment loss in imino NMR proton signal, indicating rG4 destabilization (Supplementary Figs. 8, 10e). Though these findings correlate with the atomistic MD simulation results, the simulation appears to overrepresent the structural distortion in solution as compared to the experimental results. This is evidenced by the fact that, all-atom MD simulation of the TERRA10 rG4 in isolation in 150 mM KCl solution presented a slight distortion in rG4 structure in the presence of K$^+$ ions (Supplementary Fig. 10a). The stronger structural distortion seen in our MD simulation as compared to our CD results could thus be contributed by multiple factors. The partition of K$^+$ ion into the tetrad-quartets similar to as has previously been observed for the G4s[80] accompanied by K$^+$ ion dissociation upon SERF2 binding. Alternatively, it could be an effect of the protein-RNA force field we utilized, as force field effects on G4 folding have been previously demonstrated[78,81].

## Mutational study reveals the multivalent interactions that drive SERF2 phase transition

MD simulations and NMR experiments provide the evidence that many of the evolutionary conserved lysine residues in SERF2 spanning the region 16–25 are important in rG4 binding. To further experimentally validate that these residues are involved in binding, we performed mutagenesis studies on the conserved K16, K17, K23, G24, and K25 residues that are predicted in our MD simulation to lie at the binding interfaces. Gel-shift assays show that single mutations (K23A or K25A) minimally affect rG4 binding, but the double K16-K17 mutation significantly impairs TERRA23 rG4 substrate binding (Fig. 3i). Interestingly, this same K16-K17 mutation in SERF2 also fails to bind its amyloid substrate[82]. Mutations disrupting another interface in SERF2 (K23, G24, and K25) also weaken rG4 binding, while mutating all conserved lysine residues in both binding interfaces further reduces rG4 binding. Mutations in the C-terminal Q46, K47 residues show no effect on binding as evidenced by chemical shift perturbations in the NMR experiments, indicating C-terminal contacts have minimal effects on rG4 recognition. Under crowding conditions, LLPT is slightly affected by single-point mutations (K23A or K25A) or disruption of single-binding interfaces (K16-K17 or K23-G24-K25), but LLPT is impacted when both binding interfaces are disrupted (Supplementary Fig. 10f). These findings suggest SERF2 uses multivalent interactions via multiple interfaces to drive phase separation.

## Structural insights into ribonucleoprotein condensate formation

Biomolecular condensation underpins key cellular processes and diseases[83], yet atomic-level understanding of this process is limited, especially for RNA-protein condensates. Using NMR and all-atom MD simulations, we have uncovered the detailed interactions that occur between SERF2 and TERRA rG4 under non-crowded conditions. To overcome NMR limitations that occur in crowded systems, which tend to broaden NMR signals, we added the molecular crowder PEG to our all-atom MD simulations to investigate phase separation under more physiologically relevant conditions. The simulation system contains 30 SERF2 and 30 TERRA10 rG4 molecules. Our all-atom MD simulations are predicated on the assumption that the type of interactions between protein and RNA does not significantly change upon the addition of crowding agents. Condensation is facilitated by an increase in the effective interaction network density under molecular crowding, which enhances multivalent interactions without altering the intrinsic valency of the individual components[84].

Simulation results revealed a spectrum of assemblies, from unbound TERRA10 rG4 to SERF2-TERRA10 rG4 oligomers, culminating in a dominant ring-like higher-order structure enclosing a hydrated core (~6 nm) (Fig. 4a–c, Supplementary Fig. 11a and Supplementary Movie 1). The lower-ordered 2:1 SERF2-TERRA10 rG4 complex (dilute-phase oligomers) retained a four-site binding state (Fig. 4c, top) involving N-terminal residues M1-R3-R7, R11, K16-K18, and K25-R26-R27, consistent with what was observed in a non-crowded system (Fig. 3h), suggesting early-phase continuity in interaction networks. The 2:2 SERF2:rG4 oligomers added C-terminal contacts, forming multivalent networks pivotal for phase separation (Supplementary Fig. 11b).

The ring-structure emerges from electrostatically driven clustering (Supplementary Fig. 11c) of the anionic rG4 molecules, which are bridged by multivalent, positively charged SERF2 molecules. While one might expect homogeneous mixing in condensates, the dynamic wrapping of SERF2 around clustered rG4s results in the emergence of a peripheral protein-RNA shell and a solvent-rich core (Fig. 4b, d). This architecture is multivalent in nature, within which individual SERF2 molecules bridge up to three different rG4 molecules through charged residues (Fig. 4b), creating a dynamic multivalent network that likely favors condensation. The presence of a hydrated core may reflect the anisotropic nature of RNA-protein interactions[85], where multivalent bridging and excluded volume effects combine to favor peripheral assembly.

The crowding molecule polyethylene glycol (PEG) plays an active but indirect role in this process. Initially distributed throughout the system, PEG molecules gradually become excluded from the dense core region of the condensate as it matures, likely due to steric hindrance and limited PEG permeability through the RNA-protein network (Fig. 4d, e). However, PEG is not completely excluded; transient PEG-biomolecule interactions are evident, particularly at the condensate periphery, suggesting that PEG may be playing a stabilizing role (Fig. 4e). This behavior supports PEG's role as a modulator of phase separation acting mainly by enhancing molecular crowding and interaction valency, rather than acting mostly as a direct binding partner.

In the higher-order ring-like structure, SERF2 engages multiple TERRA10 rG4s via extensive charged interfaces (Fig. 4a–c). Contact map analysis results show the N-terminal charged residues in SERF2 predominantly form contacts with TERRA10 rG4 in the large system (Supplementary Fig. 11d). A single SERF2 molecule interacts with three TERRA10 rG4s through a large interface formed by residues R3, K16, K17, K23, and R27, while smaller interfaces are involved with two

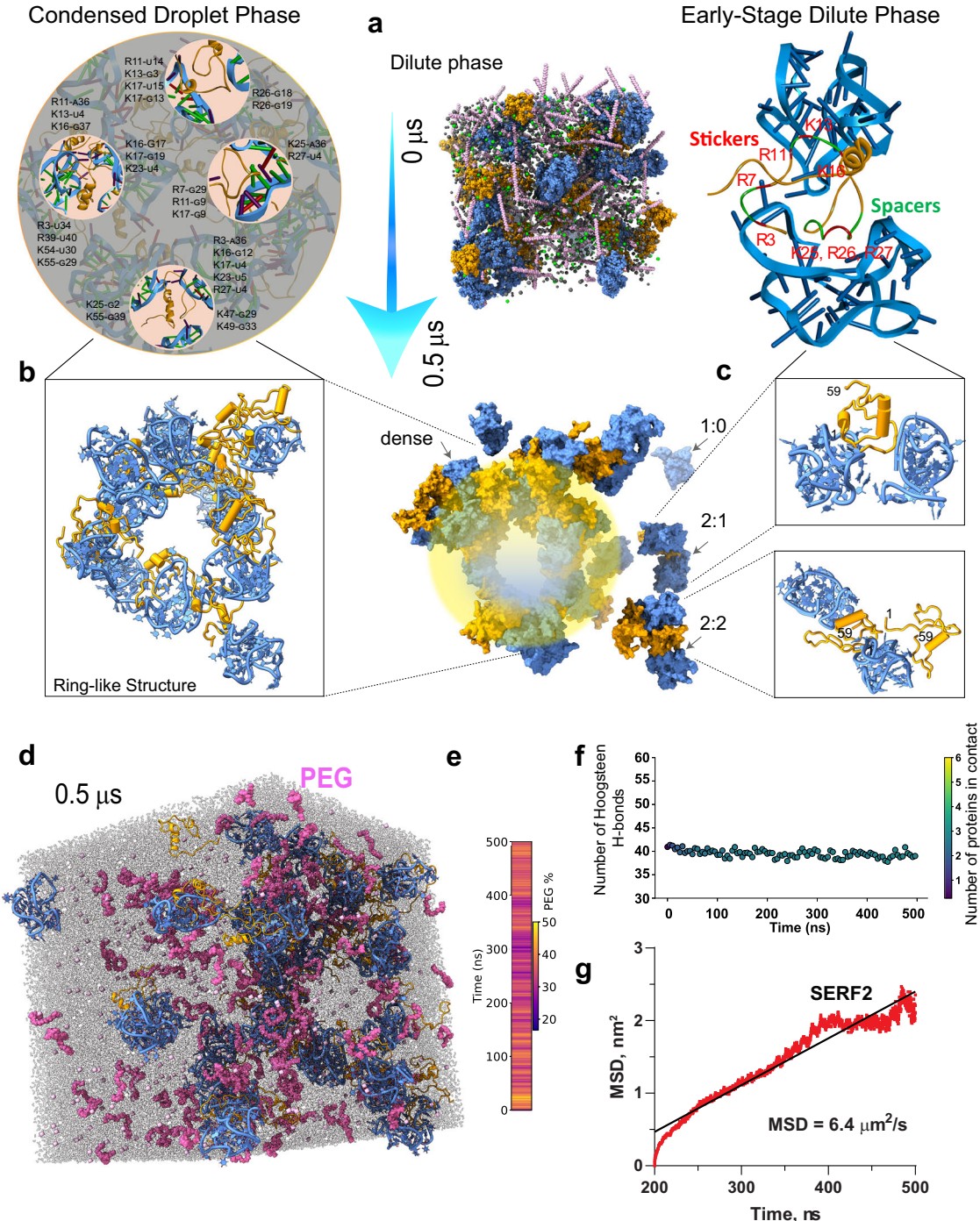

**Fig. 4 | All-atom MD simulation approach to study the structure of SERF2-TERRA rG4 phase-separated condensate. a** A cubic all-atom MD simulation box encapsulating randomly distributed 30 molecules of SERF2 (orange), 30 molecules of the TERRA10 rG4 (PDB ID: 2M18, blue), molecules of PEG are shown in pink, Cl⁻ in green, and K⁺ in gray. **b, c** Surface representation of the structure of SERF2-TERRA rG4 ring-shaped droplet-like structure found in the condensed phase (**b**), and lower-ordered oligomers (**c**), that were obtained at time 0.5 μs in the MD simulation. The enlarged all-atom cartoon structures of the condensed droplet-phase and the lower-ordered 1:2 SERF2:TERRA rG4 dilute-phase oligomers are shown on the top. The three distinct contact sites in SERF2 in the lower-ordered complex structure (**c**, top) are shown, and the TERRA rG4 interacting SERF2 residues are labeled. The high-resolution images in (**b**, top) show a representative interaction network that involves the three critical binding sites in SERF2 located in the disordered N-terminus. The SERF2 interacting residues are represented with an uppercase letter and TERRA rG4 nucleotide in a smaller-case letter (e.g., R11-U14 denotes Arg3

and uracil 14 in SERF2 and TERRA rG4, respectively). **d** Representative all-atom MD snapshot at 0.5 μs reveals the emergence of a droplet-like condensate featuring a ring-shaped assembly of SERF2 (orange) and TERRA rG4 (blue) complexes, stabilized in the presence of PEG crowding agents (magenta). **e** Quantitative analysis of PEG localization reveals time-dependent enrichment of PEG molecules within 6 Å of either SERF2 or TERRA rG4 over the course of the 0.5 μs simulation. **f** Time-resolved quantification of Hoogsteen hydrogen bonds within the TERRA rG4 structure confirms high structural integrity of the quadruplex core during the entire 0.5 μs trajectory. Color bar on the right indicates the number of protein molecules simultaneously in contact with the rG4, demonstrating that even under high protein association, the G4 core remains structurally preserved. **g** Mean squared displacement or MSD analysis reveals diffusion of SERF2 in the condensates. The 3D diffusion constant (D) extracted from the linear regime of MSD (t) using equation $D = \frac{MSD(t)}{6t}$ is calculated to be 1.07 μm²/s.

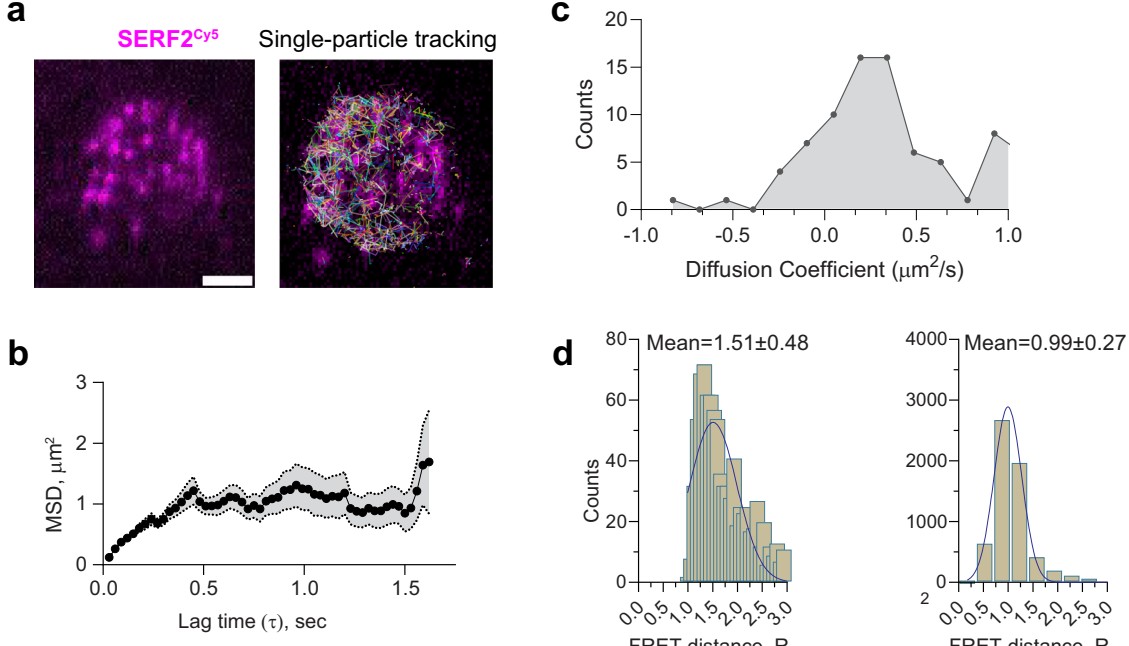

**Fig. 5 | Single-molecule imaging and tracking of SERF2 interactions with TERRA rG4s. a** Single-molecule fluorescence microscopy shows a single droplet of 50 μM SERF2 (spiked with picomolar Cy5-labeled SERF2) mixed with equimolar TERRA12 rG4s (left panel). Scale bar is 3.2 μm. Similar results were obtained in three independent repeated experiments. The corresponding single-particle tracking image of SERF2 molecules within the single droplet generated using ImageJ is shown on the right. **b** MSD plot of SERF2 (average MSD ≈ 0.95 ± 0.23 μm²) obtained from a single-particle tracking fluorescence microscopy experiment. The shaded area represents the standard error across tracked particles. Data are presented as mean values ± standard error. **c** Distribution of diffusion coefficients (D) extracted from tracking trajectories of individual SERF2-Cy5 molecules inside droplets shows a skewed and heterogeneous mobility profile centered around D ≈ 0.4–0.6 μm²/s, consistent with prior reports of protein diffusion in crowded, phase-separated compartments[90,91]. **d** FRET distance histogram between donor and acceptor was determined based on the FRET efficiency in SERF2 (T2C and A51C) and TERRA12 rG4 phase-separated droplets containing picomolar Cy3-Cy5-labeled SERF2 in dilute-phase (left) and condensed-phase (right) samples separated by centrifugation (see "Methods").

other rG4 molecules through residues K25, K55 and K47, K49 (Fig. 4b, top). Similarly, two SERF2 molecules are shown to bind two TERRA10 rG4s in a fashion where one tightly engages its N- and C-terminus, while the other shows a weak contact map through region R11-K16 and exposes its N- and C-terminus hinting at likely recruitment of additional TERRA10 rG4 partners (Supplementary Fig. 11b). This multivalent interaction pattern allows the dynamic recruitment of rG4 and protein molecules, as observed in the dilute-phase 2:2 complex (Fig. 4c). Interaction network analysis (see "Methods") showed that the charged residues are predominantly engaged in rG4 interaction and their bonding occupancy and multivalency increases as the system size increases (Supplementary Fig. 11e). In the crowded state, most SERF2 molecules bind two TERRA10 rG4s, forming the distinct ring-like structure. This ring-like structure was also observed at early simulation times and in a smaller system comprising only six SERF2 and TERRA10 rG4 molecules simulated for 1 μs (Supplementary Fig. 12a). To investigate whether the formation of this ring-like structure depends on the stoichiometry, we analyzed systems with 1:2 and 2:1 protein to RNA ratios in addition to the 2:2 ratio tested above. Our results show that the ring-like structure observed in 1:1 large system (Supplementary Fig. 12b) is also maintained in the 1:2 protein to RNA system (Fig. S12c). The 2:1 system generated a partial ring-shaped elongated arc-shaped structure similar to one observed in the 1:1 small system (Supplementary Fig. 12d). These results suggest SERF2-rG4 interactions in crowding conditions have a tendency to form ring-like structures that are maintained at different protein to RNA ratios. We note that the hollow-core feature observed here is a simulation-based observation and has not yet been experimentally validated. However, it is conceptually consistent with the sticker-and-spacer model of disordered protein phase separation, where

interaction hotspots (stickers) drive association and flexible regions (spacers) permit structural plasticity[86].

The structural stability of TERRA10 rG4 molecules in the condensates were next analyzed by calculating the formation of Hoogsteen hydrogen bonds within the rG4 molecules, with guanine N1, N2 and O2 serving as donor atoms and N7 and O6 serving as acceptor atoms, as a function of simulations time (Fig. 4f). The results showed an average ~40 Hoogsteen hydrogen bonds for the 30 rG4 molecules, very similar to a ~40 Hoogsteen bonds present within a tetrameric five-tetrad TERRA10 rG4 in isolation. These results suggested that the structural integrity of rG4 remains largely unaffected in the simulation. These results are distinct as compared to the non-crowding simulation, which showed rG4 structure distortion upon SERF2 interaction. The structural stability of TERRA10 rG4 we observe in crowding conditions could be contributed by the crowding agents, as molecular crowders are shown to enhance the stabilization of G4 structures[87]. Further, contact analysis presented an average of ~3 SERF2 molecules are in contact (within 8 Å) with one rG4 molecule (Fig. 4f).

## Single-molecule tracking of SERF2-RNA G-quadruplex liquid-liquid phase transition droplets

To validate our MD simulations, we compared the diffusion characteristics of SERF2 in SERF2-rG4s condensates obtained from MD with single-particle tracking using single-molecule fluorescence experiments (Fig. 5a). We hypothesize that the in silico computed diffusion coefficients of SERF2 molecules in condensates likely correlate to experimentally derived diffusion constants, as has been previously demonstrated for protein-only condensates[88,89]. The diffusion coefficients of SERF2 molecules in the condensed structure were computed from the all-atom MD simulation by analyzing their mean-square

displacement (MSD) (Fig.4g). These analyses yielded an average 3D diffusion constant of ~1.07 $\mu m^2/s$ (average MSD ~ 6.4 $\mu m^2$) for SERF2, indicating that the protein molecules remain dynamically mobile within the condensate environment. In contrast, diffusion in the dilute phase, primarily involving dimers and small oligomers, was significantly faster, with diffusion constants ranging from ~20–40 $\mu m^2/s$ (Supplementary Fig. 12e), suggesting rapid diffusion in these states. In contrast, the MSD of SERF2 molecules in higher-order condensates was much slower, with diffusion constants ranging from ~0.3 to 5 $\mu m^2/s$ (Supplementary Fig. 12e). This corresponds to an average experimental MSD value of $0.95 \pm 0.23\,\mu m^2$, measured in SERF-TERRA12 rG4 condensates using single-particle tracking (Fig. 5b), confirming that SERF2 molecules experience slower diffusion in the condensed phase as compared to in the dilute phase. These experimentally measured and simulation-calculated diffusion rates in SERF2-rG4 droplets (Fig. 5c) are also consistent with the ~0.1–0.6 $\mu m^2/s$ diffusion rates that have been observed for other biomolecular nucleoprotein condensates[90,91].

The slow diffusion of SERF2 in TERRA12 rG4 condensates may be due to strong electrostatic interactions of charged lysine and arginine residues with negatively charged nucleic acid bases, as has been observed for other nucleic acid-binding proteins[90]. The stability of SERF2-TERRA12 rG4 droplets even under non-physiological salt concentrations (Supplementary Fig. 13a) suggests that a robust electrostatic network exists between SERF2 and rG4. This is consistent with findings for other ribonucleoprotein droplets[90], which generally exhibit more viscous and rigid properties as compared to protein-only condensates. Indeed, the fusion time of SERF2-TERRA12 or TERRA23 rG4 droplets is very similar, as measured by dual-trap optical tweezers, which was ~2–4 s. This is one to two orders of magnitude slower than fusion rates observed in protein-only condensates[88,89] (Supplementary Fig. 13b). Additionally, FRAP revealed a relatively slow recovery half-time of ~2 min for SERF2-TERRA12 (Supplementary Fig. 13c) as compared to that observed in protein-only condensates, which generally FRAP on the seconds timescale.

The end-to-end distance between the N- (T2C) and C- (A51C) termini of SERF2 was calculated using static fluorescence resonance energy transfer (FRET) measurements in both dilute and condensed phases. In the dilute phase, the mean distance between the N- and C-termini was $1.51 \pm 0.5$ nm (Fig. 5d), consistent with NMR data suggesting close proximity between the termini in solution[39]. In the condensed phase, the mean diameter of SERF2 was $0.99 \pm 0.27$ nm (Fig. 5d), indicating that SERF2 adopts a more compact structure within the condensates. We note that, considering the experimental limitations (see "Methods") and assuming a random orientation of SERF2 within the condensate, an accurate distance estimation is not possible between a donor and acceptor. MD simulation results showed the mean distance between T2C and A51 in the condensed structure was slightly larger at $2.09 \pm 0.2$ nm as compared to the experimental results (Supplementary Fig. 13d). Single-molecule distance analysis in the MD simulation however presented several SERF2 molecules having an end-to-end distance within a range of ~1.1 to 1.3 nm (Supplementary Fig. 13d), consistent with the FRET data. We conclude that SERF2 retains a compact structure within the ribonucleoprotein condensates.

## SERF2 regulates stress granule formation and liquid phase dynamics

To further investigate SERF2's involvement in RNA binding and in LLPT, we next explored SERF2's cellular localization, function, and phase separation behavior in vivo. Immunofluorescence analysis in U2OS cells showed that under non-stress conditions, SERF2 is distributed in the cytoplasm but most intensely in the nucleolus as judged by its colocalization with the nucleolar marker protein fibrillarin (Fig. 6a, b). Interestingly, upon exposure to various stressors that include oxidative, osmotic, endoplasmic reticulum (ER),

mitochondrial, or proteasomal stress, SERF2 is primarily localized to stress granules in the cytosol as evidenced by its colocalization with five different stress granule marker proteins (Fig. 6c, d and Supplementary Fig. 14a). Nucleolar size and shape were not affected by SERF2 depletion (Supplementary Fig. 14b). We found some SERF2 colocalization with rG4 structures under stress conditions, as demonstrated by G4-specific BG4 antibody staining (Fig. 6e, f). Although rG4s are distributed across multiple cellular compartments, their enrichment in stress granules has been demonstrated recently[16,18,38]. Thus, SERF2's specific interaction with rG4s in vitro suggests a functional interplay between SERF2 and rG4s may also exist in vivo.

The depletion of SERF2 using siRNA in U2OS and BJ fibroblast cells markedly reduced the number of stress granules observed, the number of stress granule-positive cells dropped from ~94% to 20% upon sodium arsenite treatment (Fig. 7a, b, and Supplementary Fig. 15). SERF2-depleted HeLa Kyoto live cells expressing EFGP-FUS or SERF2-null HEK293T cells also showed markedly fewer stress granules than cells expressing normal levels of SERF2 (Figs. 1, 7 and Supplementary Fig. 15). The G3BP1 and G3BP2 proteins are thought to play a key role in stress granule formation[92]. We find that SERF2 depletion alone was as impactful as the combined knockdown of G3BP1 and G3BP2[93] in reducing the number of EGFP-FUS granules (Supplementary Fig. 15). In addition to contributing to stress granule formation, we found that SERF2 also influences their fluidity. FRAP measurements revealed significant fluorescence recovery occurs in about the 60 s timescale in the EGFP-FUS stress granules formed in response to sorbitol treatment control HeLa Kyoto live cells, while granules formed in SERF2-depleted HeLa Kyoto live cells showed very little recovery within this time frame (Fig. 7e). A similar, although slightly less dramatic result is seen for stress granules formed in response to sodium arsenite treatment (Fig. 7f). These results indicate that SERF2 is not only impacting the formation, size, and distribution of stress granules, but also their liquid properties. Our data suggests that SERF2 promotes stress granule formation in vivo and may enhance their fluidity likely through LLPT with rG4, as we observed in vitro (Fig. 2b). We conclude from these observations that SERF2-containing condensates are liquid-like in vitro and in vivo, unlike gels or aggregates, which are characterized by extremely slow or no measurable diffusion and exchange[94].

## SERF2 and RNA G-quadruplexes modulate the G3BP1-RNA phase transitions

Our earlier findings, which indicated SERF2 depletion could impact stress granule fluidity, motivated us to investigate whether SERF2 influences the phase behavior of the key stress granule protein G3BP1[95] in vitro. We observed that SERF2 condensates are colocalized with G3BP1 and promotes G3BP1-RNA condensation, with condensates of size up to 4.2 $\mu m$ diameter forming in the presence of SERF2, G3BP1 and total cellular RNA (Fig. 8a, b, Supplementary Fig. 16a). Notably, rG4s or short poly RNA do not induce G3BP1 LLPT in the absence of SERF2 in a non-crowding environment even when they are present at high concentrations (100 $\mu M$). In contrast, under crowding conditions (2.5% w/v PEG) where G3BP1 does not phase separate, rG4s combined with SERF2 do trigger G3BP1 condensation even at low (20 $\mu M$) protein concentrations (Fig. 8b, c, Supplementary Fig. 16b). FRAP analysis revealed that G3BP1 condensates formed in crowding conditions have a slower recovery half-time of ~450 s. On the contrary, G3BP1 showed a fast ~13-fold (34 s) half-time recovery in condensates containing SERF2, indicative of enhanced dynamics and fluidity (Fig. 8d, e, Supplementary Fig. 16c). The fast recovery of G3BP1 is also observed in condensates containing SERF2 and rG4 mixture with recovery half-time of ~67 s, but slowed to around 360 s in the presence HeLa cell total RNA. These observations suggest SERF2 and rG4 structures enhance G3BP1 dynamics and fluidity within these tricomponent condensates. SERF2 recovery was also enhanced (half-time ~10 s) in these tricomponent

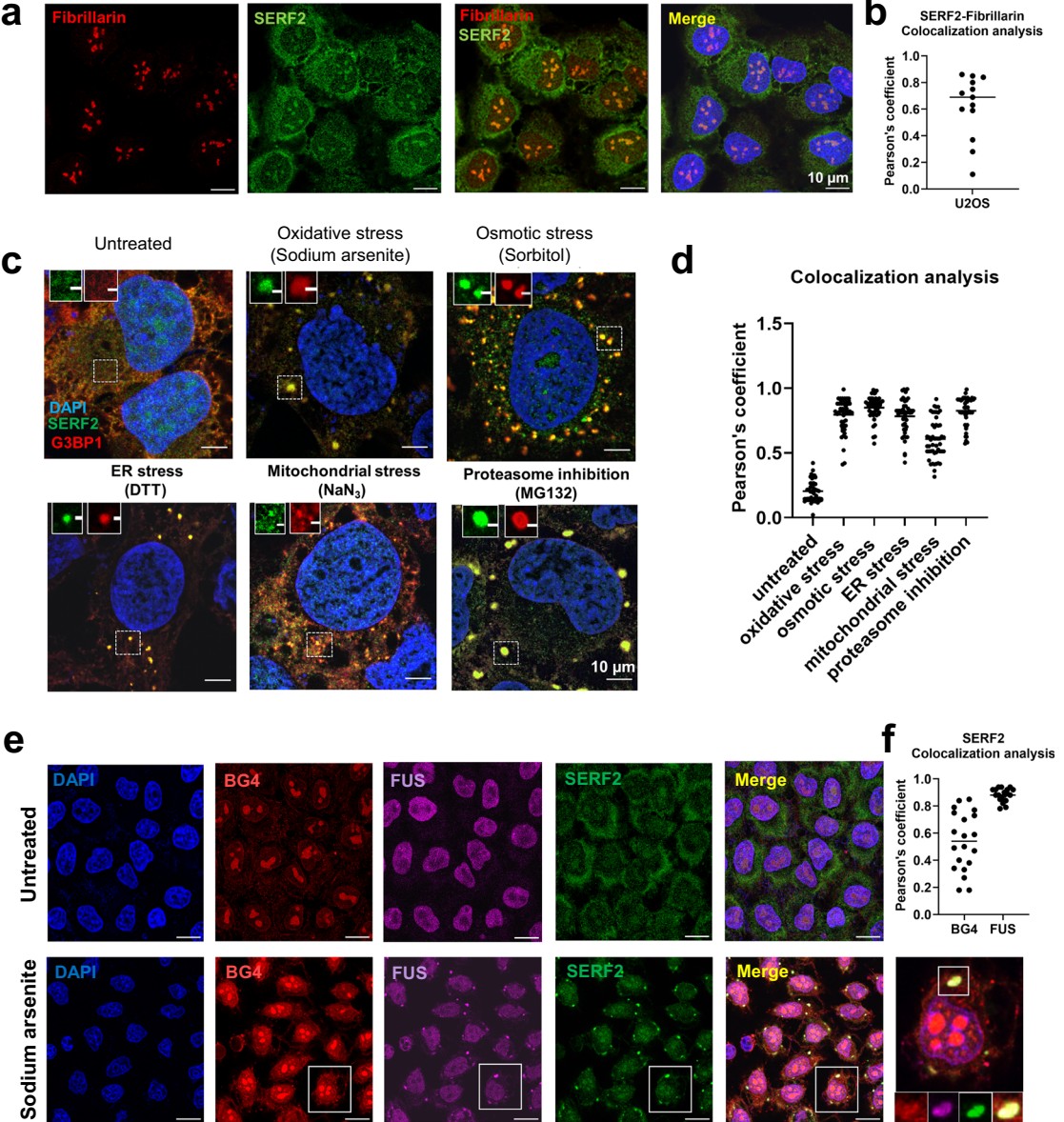

**Fig. 6 | SERF2 colocalizes with stress granules upon various stress conditions.**
**a** Immunofluorescence images show that endogenous SERF2 is predominantly distributed in the nucleolus of fixed U2OS cells, as evidenced by staining with the nucleolar marker fibrillarin. **b** Colocalization analysis of fibrillin and SERF2 obtained from (**a**). Thirteen foci from nine cells in three biological replicates were subjected to Pearson's correlation coefficient analysis. **c** SERF2 forms cytoplasmic foci and colocalizes with the core stress granule marker protein G3BP1 in different stress conditions. **d** This plot shows the quantification of stress granules retrieved from (**c**) under various stress conditions containing both SERF2 and G3BP1. At least

twenty-six foci from three biological replicates were subjected to Pearson's correlation coefficient analysis. **e** Fixed U2OS immunofluorescence cell images showing the DAPI-stained nucleus (blue) and oxidative (sodium arsenite) stress-induced granules containing G4s(red), FUS (purple) and SERF2 (green) as detected by the BG4 antibody and the FUS and SERF2 proteins. **f** Plot showing SERF2 colocalization with FUS and G4s retrieved from (**e**) as measured by the Pearson's coefficient. Twenty foci from three biological replicates were analyzed. The scale bars in (**a**, **c**, and **e**) indicate 10 μm.

condensate mixtures containing G3BP1 and rG4 as compared to those condensates that are formed in the absence of G3BP1 which show recovery halftimes of ~50–80 s (Figs. 2e, 8e, Supplementary Fig. 3g). This ability of SERF2 or G3BP1 to enhance the condensate fluidity in combination with rG4 aligns with the dynamic nature of stress granules measured in vivo, where SERF2 depletion significantly reduces fluidity of the few remaining granules (Fig. 7e, f).

## Discussion

There is a large range of liquid-liquid compartments in the cell. Though the precise function of this form of compartmentalization in health and diseases is unclear[65,96–100], one common feature is that they act to

concentrate the molecules present within them. Perhaps the most well-characterized of these compartments is the nucleolus, which is important for an efficient ribosome assembly[101]. Another well-characterized compartment is the stress granule, which may be involved in RNA or protein storage during stress[97]. Many of the proteins within liquid-liquid phase compartments contain elements of disorder, and understanding how they function and interact with other molecules is limited. This is due to a lack of experimentally determined conformational ensembles for disordered proteins, making it difficult to study their interaction with RNA species. We have a broad overview of factors affecting LLPT, including the importance of multivalency and disorder. However, the lack of detailed structural information

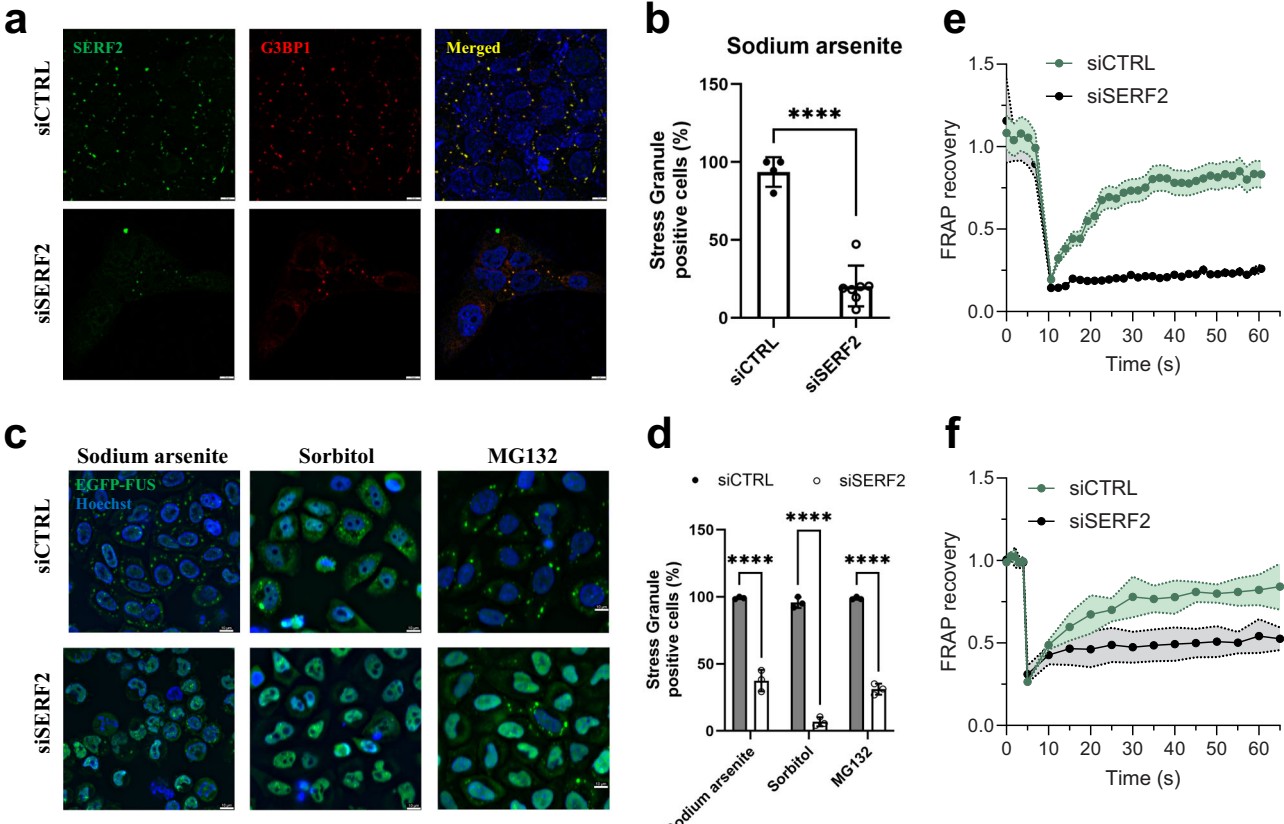

**Fig. 7 | SERF2 regulates stress granule formation and dynamics.**
**a** Immunofluorescence of SERF2 and G3BP1 in fixed U2OS cells treated with 0.5 mM sodium arsenite for 1 h. **b** Plot shows the percentage of stress granule-positive cells after sodium arsenite treatment calculated from images shown in (**a**). Error bars were calculated from four independent experiments, and data are presented as mean values. **c** Live-cell imaging of EGFP-FUS HeLa Kyoto cells treated either with a control RNA (siCTRL) or an RNA targeting SERF (siSERF2), treated with different stressors (0.5 mM Sodium arsenite, 0.4 M Sorbitol, or 10 μM MG132) for 1 h. Scale bars in (**a** and **c**) are 10 μm. **d** Plot shows percentage of stress granule-positive cells under different stress treatments calculated from images shown in (**c**). At least 50 cells from three biological replicates were analyzed, and data are presented as mean values. **** shown in (**b** and **d**) indicates $P < 0.0001$. **e, f** These graphs show FRAP recovery curves in EGFP-FUS HeLa Kyoto live cells with siCTRL or siSERF2 conditions treated with 0.4 M sorbitol (**e**) and 0.5 mM sodium arsenite (**f**). Standard deviations were calculated by analyzing four different foci in three replicates subjected to FRAP, and data are presented as mean values.

leaves both the protein and RNA components of these membraneless compartments often diagrammed as vague spaghetti-like lines. A handful of structural models for condensates have been reported recently[88,89], but these have been limited to protein-only condensates, and our understanding of ribonucleoprotein condensates remains largely unexplored. Our replacing these vague lines with the structural and biophysical details diagrammed in Figs. 3 and 4 helps us understand the process of LLPT and the interactions that take place within these cellular compartments.

Here we have obtained detailed structural information for two important components of stress granules, namely SERF2 and rG4, both separately and in combination, helping us to understand the process of LLPT and the interactions that likely take place within these membraneless compartments. We show that SERF2,[37]is a component of the nucleolus and stress granules and appears to be important for stress granule formation in vivo. We also show that SERF2 specifically binds to rG4s in high-throughput and in vitro assays. These non-canonical RNA structures have been implicated in translational repression and transcriptional termination[19], mRNA processing, telomere maintenance, RNA translocation and can effectively influence protein aggregation[19,47,51,102–105]. In findings likely relevant to ours, it has recently been found that stress enhances the number of rG4 present[17]. rG4 are abundant in stress granules, and stress promotes rG4 folding[17]. SERF2 binds to known rG4 structures, including some that are known to be present in stress granules, with sub-micromolar binding affinities. As

SERF2 and TERRA rG4 both in isolation are shown to promote amyloid formation[47,106], we speculate that under unfavorable conditions like stress, their relatively strong interactions could suppress their ability to bind and promote aggregation of amyloid substrates, a phenomenon we demonstrated for the SERF2 homologous protein Znf706[27].

Despite the lack of direct intermolecular NOEs or side-chain-specific assignments, which remain a major technical challenge in NMR studies of ligand or dynamic protein-rG4 complexes[107], our integrative approach combining chemical shift perturbations guided docking and MD simulations offers a structurally and biophysically plausible model[108–111]. This strategy reflects current best practices for mapping challenging or transient interactions involving intrinsically disordered regions and provides experimentally testable hypotheses that we further validated through mutagenesis. Our studies on SERF2 and TERRA rG4 provide a structural framework for understanding protein-RNA condensates and how multiple transient electrostatic interactions can drive oligomer formation. Upon interaction of SERF2 with rG4, a planar G-quartet RNA-protein interaction forms through a quadrupole-like interaction. The SERF2 protein dynamics are constrained upon rG4 binding, and the rG4 structure becomes distorted (Fig. 3). The conformational adaptability of SERF2 enables it to interact with the TERRA rG4 structure, which induces destabilization in the TERRA rG4 structure, indicative of SERF2 being involved in rG4 unfolding in non-crowding conditions. Several RNA-binding proteins, such as FMRP, nucleolin, CNBP, eIF4A, hnRNPA1, and DHX36, have also been

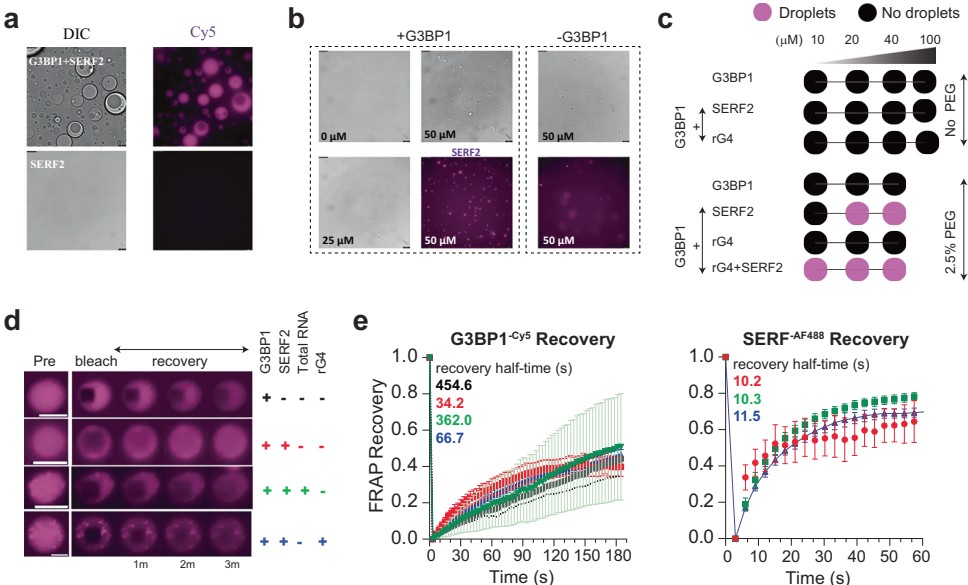

**Fig. 8 | SERF2 facilitates the phase transition of the stress granule core protein G3BP1 and enhances G3BP1 fluidity. a** DIC and fluorescence images showing co-phase separation of SERF2 (purple) and G3BP1 (unlabeled) with 12.5 ng/μL HeLa total RNA. Similar results were obtained in three independent repeated experiments. **b** DIC and fluorescence images show SERF2 facilitates G3BP1-RNA condensation in samples containing the indicated SERF2 and G3BP1 concentrations and 12.5 ng/μL HeLa total RNA. Similar results were obtained in three independent repeated experiments. Scale bars in (**a** and **b**) are 10 μm. **c** Phase diagram showing G3BP1 phase transition in a non-crowding condition and a crowding condition that contained 2.5% w/v PEG8000 in a 20 mM NaPi, pH 7.4, 100 mM KCl buffer. G3BP1

phase transition was measured in the absence or presence of SERF2 or rG4 or mixtures of SERF2 and rG4s as indicated. **d** Time-series images of 50 μM G3BP1 condensates after photobleaching in the absence or presence of SERF2, total RNA, and TERRA23 rG4s prepared in 20 mM NaPi, 100 mM KCl (pH 7.4) containing 5% w/v PEG8000. **e** FRAP recovery plots of G3BP1-Cy5 (left) and SERF2-AF488 (right) obtained from (**d**), the graph colors correspond to the plus and minus signs of the sample mixture shown in (**d**). Recovery half-time is calculated by non-linear regression fit and one-phase association using GraphPad Prism. Scale bar is 5 μm. Standard deviations were calculated by analyzing 8 isolated droplets subjected to FRAP, and data are presented as mean values.

shown to bind rG4 and modulate their folding[20–24,26]. Such interactions seem to be able to regulate their cellular functions, for instance, yeast Rap1 interacts with G4 using planar G-quartets to stabilize the complex[112]. In contrast, the DHX36 helicase unfolds G4[31] by forming a flat non-polar surface on the G-quartet[101], in a way similar to that observed for SERF2 interacting with TERRA rG4. However, under macromolecular crowding conditions, which mimic the dense intracellular environment, the TERRA rG4 structure remains stable even in the presence of SERF2. This suggests that the crowded milieu promotes and stabilizes rG4 folding, an effect consistent with known biophysical principles[113]. Crowding has been shown to favor compact nucleic acid structures, including G4s, by shifting the equilibrium toward folded states due to excluded volume effects and dehydration of the RNA backbone[87,113,114]. These environmental constraints may counteract SERF2-induced destabilization, thus preserving rG4 architecture despite protein binding. Our data therefore suggest that SERF2 may function as an environment-dependent modulator of rG4 structure, where it is capable of perturbing rG4 conformation in dilute environments, but its destabilizing influence is diminished or buffered in crowded conditions. This suggests a potential regulatory switch wherein SERF2's impact on RNA structure may be tightly controlled by the physicochemical environment, such as that found within stress granules.

The pattern of charged residues in disordered proteins and the charge interactions have been demonstrated to play an important role in LLPT, as described in the simplified stickers and spacers model[86]. Excitingly, we can now start to define stickers at an atomic level based on the frequency with which specific protein residues appear in our condensate model to be interacting with RNA (Fig. 4). Our simulation results suggest that charge interactions and nucleic acid conformational changes may help stabilize the dynamic condensates and facilitate their selective recruitment of specific proteins and nucleic acids during phase transition. We find that distortion of nucleic acid

structures, including unwinding or bending, enhances interactions by exposing charged regions, and shielding RNA-RNA interactions, and promoting multivalent intermolecular interactions may be important for phase separation as previously predicted[99]. Unexpectedly, we frequently observed ring-like structures that bring multiple SERF2 and TERRA rG4 molecules together through multivalent interactions. In support of a physiological role for macromolecular crowding, we find that the PEG crowding agent becomes partially excluded from the ring-like condensates while being involved in interactions at the periphery of the rings. As this is the only structure that we are aware of a liquid-liquid phase-separated protein-RNA complex, we have no way of knowing how common these ring-like structures will prove to be in phase-separated complexes. However, their discrete size represents an attractive mode of interaction that could act to maintain an appropriate degree of exchangeability of the condensate components while avoiding higher-order interactions that could form insoluble complexes like aggregates.

Though the current work provided structural insights into SERF2 binding to TERRA rG4 and how it distorts the rG4 structure, future work will be necessary to probe if it interacts similarly or differently with in vivo rG4 binding partners, in stress granules or elsewhere. Determining if the ring-shaped structures we have observed are a common feature within condensates likely awaits the development of other protein-RNA pairs that can be analyzed in the same detail as our SERF2-rG4 system. A better understanding of the structure-function relationships present within quadruplexes will aid us in understanding rG4-linked biological functions such as gene regulation and stress granule formation.

## Methods

### Cell culture, treatment, and transfection

U2OS cells (human) were purchased from Sigma-Aldrich (Cat. no. 92022711-1VL), and HEK 293T/17 cells (ATCC® CRL-11268™) used in this

study are derived from human embryonic kidney tissue. The HeLa Kyoto eGFP-FUS cells (human) were obtained from Dr. Antony Hyman at Max Planck Institute of Molecular Cell Biology and Genetics, Germany. All cells were cultured in Dulbecco's Modified Eagle Medium (Fisher Scientific, 11-995-073) supplemented with 10% heat-inactivated fetal bovine serum (Sigma, F4135) and 1X Penicillin-Streptomycin-Glutamine (Fisher Scientific, 10-378-016). Stress treatments were performed on cells with vehicles alone or with different stress inducers. Cells were then treated with a final concentration of 0.4 M sorbitol or 0.5 mM sodium arsenite to induce osmotic or oxidative stress, respectively. For ER stress, the cells were additionally treated with a final concentration of 2 mM dithiothreitol for 30 min in a culture medium at 37 °C. Cells were treated with 10 μM of the proteasome inhibitor MG132 (Cas No- 133407-82-6, Sigma) diluted in culture medium and incubated for 30 min at 37 °C to inhibit protease function. For mitochondrial stress, cells were treated with 75 mM $NaN_3$ for 30 min at 37 °C. For heat shock, cells were incubated at 43 °C for 1 h. For plasmid transfection, cells were transiently transfected using Lipofectamine™ LTX and PLUS™ reagent (Fisher Scientific, 15338030) according to the manufacturer's instructions. For knockdown, 13 nM of control (Horizon Discovery, D-001206-13-20) or SERF2 (Horizon Discovery, M-016317-01-0010) siRNAs were transfected using RNAi-MAX reagent (Thermo Fisher, 13778150). After 48 h of incubation, the transfected cells were harvested for western blot analysis and RT-qPCR.

## Immunofluorescence

Cells were grown on coverslips with the conditions described above and then fixed with 4% paraformaldehyde (Electron Microscopy Sciences, 157-8-100) in 1X DPBS (Fisher Scientific, 14-190-144) at room temperature for 10 min. The fixed cells were washed three times with 1X DPBS and then permeabilized with 0.1 % Triton X-100 at room temperature for 30 min. The permeabilized cells were next blocked at room temperature for 1 h using UltraCruz™ blocking reagent (Santa Cruz, sc-516214), followed by incubation with primary antibody solutions for 1 h at room temperature. Primary antibody solutions were prepared in UltraCruz™ blocking reagent using the following dilution factors: 1:200 rabbit anti-SERF2 (Proteintech, 11691-1-AP), 1:1000 mouse anti-G3BP1 (BD Biosciences, 611127), 1:1000 mouse anti-Fibrillarin (Boster Bio, M03178-3), 1:100 mouse anti-Nopp140 (Santa Cruz, sc-374033), 1:100 mouse anti-USP10 (Santa Cruz, sc-365828), 1:200 mouse anti-FUS (Thermo Fisher, 50-554-337), 1:500 mouse anti-BG4 (Absolute antibody, Ab00174-1.1), 1:100 mouse anti-TIA1 (Santa Cruz, sc-398372), and 1:100 mouse anti-eIF2α (Santa Cruz, sc-133132). The cells were then washed three times with 1X DPBS and incubated in secondary antibody solutions in the dark at room temperature for 1 h. Secondary antibody solutions were prepared in a blocking reagent using a 1:1000 dilution, goat-anti-rabbit secondary Alexa Fluor Plus 488 (Thermo Scientific, A32731), and goat-anti-mouse secondary Alexa Fluor 647 (Thermo Scientific, A21235), goat-anti-mouse secondary Alexa Fluor 488 (Invitrogen, A11001). After three washes with 1X DPBS, the cells were stained with 0.25 μg/mL DAPI (Thermo Scientific, D1306) in 1X DPBS for 3 min. The cells were washed three more times with 1X DPBS, air dried, and mounted with ProLongGold™ antifade reagent (Cell Signaling, 9071S) on glass slides. After 24 h of mounting, the slides were sealed with nail polish and imaged using a Leica SP8 confocal or Thunder™ microscope.

## Oligonucleotide synthesis

High-performance liquid chromatography (HPLC) purified unlabeled, TAMRA- and 6-FAM fluorescent labeled RNA nucleotides were either purchased from Integrated DNA Technology (IDT) or synthesized at the Slovenian NMR Center using previously described methods[115,116]. Different length TERRA RNA sequences that include TERRA10, TERRA12, TERRA23 and TERRA60; 4-repeat GGGGCC, six-repeat

UGGGGU, short poly RNA, HIV-1 TAR RNA, CCCCGG and mutated TERRA repeat RNA were synthesized from IDT. Random length polyA was purchased from Sigma. All the oligonucleotides were suspended in nuclease-free water or buffer prepared using nuclease-free water. All oligonucleotides were desalted using a 3 kD cutoff filter (Amicon ®Ultra 0.5 mL) by treating with 20 mM Tris-HCl, pH > 11, heated at 95 °C for 3 min following 10-time buffer exchange in 20 mM Tris-HCl, pH 7.4. The oligonucleotide concentration was next measured using the extinction coefficients obtained using the IDT OligoAnalyzer™ tool. G4 sequences were folded by cooling the samples prepared in KCl (20 mM sodium phosphate (referred to as NaPi hereafter), pH 7.4 and 100 mM KCl) or LiCl buffers (20 mM Tris-HCl, pH 7.4 and 100 mM LiCl) using a thermocycler with 1 °C/min. The folded G4s were stored at 4 °C for immediate use or at −20 °C for future use. All the chemicals and reagents used in this study were commercially purchased with >98% purity and used without further purification.

## Purification of recombinant proteins

The plasmids, containing a codon-optimized human SERF2 gene, were synthesized commercially by GenScript and subcloned to a pER28a-SUMO vector as reported elsewhere[39,42]. The expression and purification of the G3BP1, SERF2 wild-type or different lysine-alanine mutant proteins was like that previously reported for yeast SERF[42]. Briefly, the plasmids were transformed into competent BL21(DE3) *Escherichia coli* cells, incubated overnight in 10 mL of cell culture medium, and then transferred to freshly prepared 1 L of PMEM medium containing 50 mg/L Kanamycin. The cells were further grown at 37 °C, under shaking, until the $OD_{600}$ reached 1.0. Then they were transferred to a 20 °C shaker for 1 h, protein expression was induced by adding 0.1 mM IPTG, and the cells were incubated overnight at 20 °C with continued shaking.

Isotope-labeled 15 N and 15N-13C SERF2 proteins for NMR studies were produced by growing cells in M-9 minimal media supplemented with 100% 15 N $NH_4Cl$ (1 g/L) for the 15 N labeling or 15 N $NH_4Cl$ and D-Glucose-$13C_6$ (4 g/L) for the 15N-13C labeling. Cells were subsequently harvested and lysed by sonication in ice-cold lysis buffer (40 mM Tris-HCl pH 8.0, 10 mM NaPi, 400 mM NaCl, 20 mM imidazole, 10% glycerol, 1 tablet of cOmplete protease inhibitor (Roche), and 1.25 μg/mL DNase I (Roche)). The lysate was centrifuged at 36,000×*g* for 30 min, and the supernatant was passed through a HisTrap column (Cytiva, 17-5248-02). Lysis buffer containing 0.5 M imidazole was used to elute the His-SUMO tagged SERF2 proteins; the elution was then supplemented with beta-mercaptoethanol to a final concentration of 5 mM. The sample was next incubated overnight at 4 °C with 10 μL of homemade SUMO protease 6His-ULP1 for His-SUMO cleavage. The digestion mixture was dialyzed in 40 mM Tris-HCl pH 8.0 and 300 mM NaCl overnight at 4 °C, using a 3.5 kD cutoff dialysis membrane (Repligen, 132724). The dialyzed proteins were run through a 5 mL HisTrap column to remove the cleaved His-ULP1 and the His-SUMO. The flow-through SERF2 protein was further purified by an ion exchange HiTrap SP column (Cytiva, 17-5161-01) using buffer A (50 mM NaPi and 125 mM NaCl pH 6.0) and buffer B (50 mM NaPi and 1 M NaCl, pH 6.0). A final purification of the ion-exchange purified SERF2 protein was conducted using a size-exclusion chromatography column Hiload75 (Cytiva, 28989333) in 20 mM NaPi pH 7.5 and 150 mM NaCl or 40 mM HEPES pH 7.5 and 100 mM NaCl. The protein samples used in the biophysical and biochemical studies were prepared in indicated buffers as needed via buffer exchange using a 3 kD cutoff filter (EMD Millipore, UFC503024). The SERF2 protein concentration was determined using a Pierce TM BCA assay calibrated with the SERF2 A51W mutant serving as a standard. A similar expression and purification protocol, as described above, was used to produce different SERF2 cysteine and alanine mutants.

For the high-throughput RNA Bind-n-seq screening assay, SERF2 protein was prepared containing GST and streptavidin binding peptide

(SBP) tags. SERF2 was cloned into a pGEX-6P-1 construct containing GST and SBP tags and transformed into Rosetta-competent cells. Cell cultures were grown in LB at 37 °C until $OD_{600}$ reached ~0.6, thereafter induced with 0.5 mM IPTG overnight at 16 °C. Cells were harvested at 4000×$g$ for 13 min and resuspended in lysis buffer (1% Triton X-100, 5 mM DTT, 4 mM $MgCl_2$, 200 mM NaCl, 20 mM HEPES and 1 tablet/L culture Pierce™ Protease Inhibitor Mini Tablets EDTA-free). Lysates were sonicated and incubated with 3 units/L culture RQ1 DNAse (Promega) and 500 units/L culture Benzonase Nuclease (Sigma-Aldrich) for 15 min at room temperature. Following incubation, lysates were centrifuged at 36,000×$g$ for 30 min, and the supernatant was passed through a 0.45 μm filter. GST-SBP-SERF2 was purified using Pierce™ Glutathione Agarose (Thermo Fisher Scientific). Protein was further cleaned up using HiTrap® Heparin High Performance (Sigma-Aldrich), and concentration was assessed using Pierce 660 nm assay (Thermo Fisher Scientific). Protein purity was determined by SDS-PAGE and Coomassie blue staining.

## Protein labeling

Cy5 (Cytiva, PA25031) or Alexa Fluor 488 (AF488) (Invitrogen, A10254) labeling of protein was done by incubating 200 μM of SERF2 (T2C) or G3BP1 with a 10-molar excess of $C_5$ maleimide mono-reactive dye in 20 mM Tris-HCl pH 7.4 and 100 mM KCl buffer overnight, at 25 °C (SERF2) or at 4 °C (G3BP1), under continuous shaking at 300 rpm. The free excess label was then removed by passing the sample through a PD-10 desalting column in a dark room. The samples were concentrated using a 3 kD Amicon Ultra-15 Centrifugal Filter Unit (EMD Millipore, UFC800324), and any residual free dyes were removed by resuspending the proteins in a working buffer after centrifugation at 7000×$g$ (8 times) for 15 min using an Amicon ®Ultra 0.5 mL 3 kD cut-off filter.

## High-throughput screening assays

*RNA bind-n-seq assay and analysis*—For the RNA bind-n-seq assay, a single-strand DNA library containing a randomized 40-nucleotide region was obtained from IDT, gel-purified, and the RNA library was prepared following a previously described method[48] using a T7 promoter and in vitro transcription. A second pool of RNA was made by replacing the guanines with 7dG (Trilink) to eliminate rG4 folding while preserving the sequence. Residual DNA was removed with DNase I (Promega), followed by a phenol-chloroform extraction. The RNA was resolved in a 6% TBE-Urea gel, the expected size band was excised, and gel purification was conducted as previously described[48].

The RNA bind-n-seq method was modified from a previous study as described below[48]. Briefly, 60 μL of recombinant GST- SERF2 at different concentrations (250 nM and 50 nM) in binding buffer (25 mM Tris-HCl pH 7.5, 150 mM KCl or LiCl, 3 mM $MgCl_2$, 500 μg/mL Ultrapure BSA and SUPERase-In RNase Inhibitor) were equilibrated with 60 μL of pre-washed (binding buffer) magnetic beads (Dynabeads MyOne Streptavidin T1, Invitrogen) for 30 min at 4 °C. RNA pools were heated in the presence of 150 mM KCl or LiCl for 5 min at 100 °C and cooled down to room temperature for at least 10 min. 60 μL of 3 RNA pools (KCl, LiCl, and 7dG (in LiCl)) were then mixed with GST-SBP-SERF2 and further incubated for 1 h at 4 °C. The final concentrations of protein in the binding reaction were 250 nM and 50 nM, and the RNA's concentration was 1 μM. The protein-RNA complexes were washed with buffer (25 mM Tris-HCl pH 7.5, 150 mM KCl or LiCl, and SUPERase-In RNase Inhibitor (Invitrogen)). The complexes were magnetically isolated, and RNA was eluted with buffer (4 mM biotin and 25 mM tris-HCl pH 7.5) for 30 min at 37 °C. The elution was performed twice, the eluates were combined, and RNA was purified by a phenol-chloroform method[48]. Before reverse transcription, RNAs were heated for 5 min at 100 °C in the presence of 150 mM LiCl to facilitate rG4 unfolding. Following reverse transcription, the samples were prepared for sequencing as described elsewhere[48].

We performed sequence enrichment analysis as previously described[48,117]. Briefly, we analyzed k-mer (k = 6) enrichments to determine 'R' values. R was defined as the frequency of a given k-mer in the protein-associated pool divided by the frequency of that *k*mer in the input pool. To specifically determine the enrichment of rG4s, we searched for strong G4 patterns ($G_{(3-6)}N_{(0-7)}$)$_4$. As a control, we removed all sequences that matched the G4 pattern but still had greater than 8 guanines in the randomized region. The G4 pattern analysis was performed on the randomized region plus the adapters (as described in ref. 48 for RNA structure analysis), as they are part of the RNAs presented to the protein in the binding reaction. The enrichment of each pattern was the frequency of the pattern in the protein-bound sample divided by its frequency in the input pool.

*FOREST assay*—Library-1 from the previously published paper[49], which includes 1800 pre-miRNA and 10 rG4 sequences, was used for FOREST screening. A detailed method for oligonucleotide template pool, DNA barcode microarray design, in vitro transcription, RNA fluorophore labeling, hybridization, and microarray scanning has been provided in the supporting information. Briefly, the templates used were synthesized by oligonucleotide library synthesis, and the size was limited to 170 nucleotides (OLS, Agilent Technologies). The in vitro transcribed RNA structure library was labeled with Cy5 at the 3' end to detect and quantify RNA probes on a microarray. Lastly, the library was prepared in $K^+$ folding buffer (10 mM Tris-HCl pH 7.5, 100 mM KCl), heated at 95 °C and cooled to 4 °C at a rate of −6 °C/s on a ProFlex Thermal Cycler (Thermo Fisher Scientific) to allow for G4 folding.

His-tagged SERF2 recombinant protein was used for the FOREST binding assay. For this purpose, the target protein (100 pmol of SERF2), 20 μL of TALON magnetic beads (Clontech), and 1 μg of the refolded RNA structure library were mixed in 1 mL of protein-binding buffer (10 mM Tris-HCl pH 7.5, 100 mM KCl, 10% glycerol, and 0.1 μg/μL BSA). A mixture containing no protein was also prepared as a control. The mixtures were incubated on a rotator at 4 °C for 30 min and washed three times with the protein-binding buffer. Then, 200 μL of elution buffer was added to the magnetic beads, and the mixture was heated at 95 °C for 3-min. The RNA was collected from the supernatant by removing the magnetic beads. The RNA structure library was extracted with phenol and chloroform, and an ethanol precipitation was conducted for RNA purification. The enriched RNA sample was hybridized for microarray analysis as detailed in the supporting methods. To determine the protein-binding intensities of each RNA probe, we subtracted the fluorescence intensities of the negative control sample (samples without protein) from those of the enriched protein samples. To account for any undesired interactions with the barcode region, we calculated the average fluorescence intensity of each structure by averaging the intensities of the RNA probes that had the same RNA structure but differing barcodes.

## NMR experiments

Unless indicated, all NMR measurements were done with protein or protein-RNA mixture samples that had been suspended in 20 mM NaPi and 100 mM KCl (pH 7.4) containing $D_2O$ at 4 °C. Isotopically unlabeled protein and oligonucleotide (TERRA10, TERRA12, and TERRA23) samples were used to collect 1D proton NMR spectra with a recycle delay of 1 s. Multidimensional NMR data were collected using uniformly isotope-labeled 15 N and 15N-13C SERF2 samples. A series of 2D and 3D NMR experiments, which included 15N-HSQC,13C-HSQC, HNCO, HNCA, HNCOCA, HNCACO, CBCA(CO)NH, HNCACB, 15N-HSQC-NOESY and 15N-HSQC-TOCSY, were conducted for backbone assignments of 1.2 mM proteins dissolved in 20 mM NaOAc-d$_3$ with pH 5.5 and 8% $D_2O$ at 4 °C. 2D 15N-1H HSQC experiments were recorded for 100 μM 15 N labeled SERF2 titrated with an increasing concentration of different length TERRA (10, 12, 23 or 60) rG4 for binding site analysis. The effect of pH and temperature on SERF2's chemical

exchange was studied by acquiring the 2D spectrum at 4 and 37 °C, as demonstrated elsewhere for disordered proteins[72], in buffers containing either 20 mM NaPi and 100 mM KCl (pH 7.4) or 20 mM $d_3$-NaOAc (pH 5.5). 15 N relaxation NMR measurements, including heteronuclear NOE, T1, and T2, were done using 200 µM 15 N SERF2 mixed with and without 100 µM of TERRA rG4 at 4 °C in 20 mM NaPi and 100 mM KCl (pH 7.4). The relaxation delays used for the T1 experiments were 20, 50, 90, 130, 200, 320, 450, 580, 750, 900, 1200, and 2005 ms. T2 relaxation delays used were 16.96, 50.88, 84.80, 135.68, 169.60, 220.48, 305.28, 373.12, 457.92, 542.72, 678.40, and 1356.80 ms. 1D proton NMR experiments were carried out to map the chemical shift changes in the TERRA rG4 imino protons (200 µM TERRA12 or 100 µM TERRA10) at increasing SERF2 concentration. Saturation transfer difference (STD) NMR spectra were recorded with 512 scans and a saturation time of 4 s. On-resonance saturation was applied at selected well-resolved chemical shift regions, and off-resonance saturation was applied at −40.0 ppm to serve as a control. NMR samples used for STD NMR data collection consisted of 100 µM TERRA12 rG4, 100 µM SERF2, or a 1:1 mixture of SERF2 and TERRA12 rG4, prepared in 20 mM NaPi, 100 mM KCl, pH 7.4. To test for direct magnetization transfer and to exclude potential contributions from solvent exchange, we selected isolated and well-resolved imino proton peaks from TERRA12 rG4 and an isolated backbone amide proton peak (-8.5 ppm) from SERF2. These peaks were first saturated in the individual samples (RNA or protein alone) with off-resonance irradiation serving as the reference. The same peaks were then saturated in the protein-RNA complex sample under identical conditions. Differences in saturation transfer between the free and mixed states were analyzed to verify that the observed transfer is due to direct protein-RNA interactions and not mediated by solvent exchange. NMR data were collected on a Bruker 800 MHz spectrometer equipped with a triple resonance cryoprobe, while 15 N relaxation data were collected on a Bruker 600 MHz NMR spectrometer equipped with a triple resonance cryoprobe. The NMR data were processed using Bruker's Topspin 4.1.4, and spectra assignment and analysis were done using an NMRFAM-Sparky 1.47.

### Fluorescence polarization and anisotropy assay

Fluorescently labeled 6-FAM RNA or AF488 labeled SERF2 probes were prepared in nuclease-free water containing 20 mM NaPi (pH 7.4) and 100 mM KCl. Fluorescence polarization assays were done either using 20 to 100 nM RNA 6-FAM rG4 or non-rG4 RNA probe mixed with increasing concentrations of SERF2, ranging from low nM to high µM, or using 200 nM AF488 labeled SERF2 with increasing concentration of structured or unstructured RNA. All fluorescence polarization (FP) assays were done in samples dissolved in 20 mM NaPi and 100 mM KCl (pH 7.4). In addition to unlabeled TAR RNA binding to AF488 SERF2, FP assay was also done using TAMRA-TAR RNA to reveal SERF2 binding specificity for rG4 structured RNA. FP or gel-shift EMSA binding assay was also carried out to test the binding specificity of SERF2 to two rG4 sequences derived from the 3' UTR regions of *Mark2* and *Stxbp5* genes. The sample mixtures for FP assay were incubated for 30 min at room temperature. Fluorescence polarization data were recorded on a TECAN Infinite M1000 microplate reader at 25 °C with the excitation/emission wavelengths set at 470/530 nm, for FAM or AF488 probes; 530/575 nm for TAMRA probes. Fluorescence anisotropy measurements were done by titrating SERF2 to 200 nM of Cy3 (excitation/emission, 550/570 nm) or FAM (excitation/emission, 493/517 nm) labeled oligonucleotides, in a 1 mL quartz cuvette (Hellma,101-QS), using a Cary Eclipse spectrofluorometer (Agilent) at 25 °C. Slit bandwidths were set at 5 nm and 10 nm for excitation and emission, respectively. The binding constant ($K_D$) was calculated from the change in polarization or anisotropy values, in GraphPad Prism 9.5.1, using non-linear regression for curve fitting with a one-site specific binding model.

### Circular dichroism spectroscopy

The secondary structure of the G4s (15 µM) or SERF2 (50 µM), suspended in 20 mM NaPi and 100 mM KCl (pH 7.4), was studied by collecting circular dichroism (CD) spectra using a JASCO J-1500 spectropolarimeter. For folding analysis, the CD spectra of SERF2 were collected at different temperatures (4, 25, and 37 °C). CD titration was done for 20 µM TERRA23 rG4 against an increasing SERF2 concentration (10 to 100 µM) to monitor rG4 secondary structure change at 25 °C. The buffered CD spectrum was subtracted from the average CD spectrum obtained from 8 scans. The structural change in 40 µM SERF2 mixed with or without 20 µM TERRA12 rG4 was measured, which corresponds to 2:1 protein to RNA ratio used in NMR relaxation experiments. CD spectra of rG4 alone were used as a baseline and subtracted from the spectra of the SERF2-rG4 complex to monitor secondary structural change in SERF2.

### In vitro liquid-liquid phase transition assay

16-well Culture-Well chamber slips (Grace Bio-Labs) or 384-well plates (Cellvis, P384-1.5H-N), pre-treated with 5% (w/v) Pluronic™ F-127 (Sigma, P2443) overnight, were used to study in vitro phase separation. The well chambers were washed three times with 20 mM NaPi, 100 mM KCl, pH 7.4 buffer and air dried. 50 or 20 µL aliquots of reaction sample mixtures were incubated for 30 min at room temperature in various conditions (varying protein, total RNA from HeLa cells, poly-ribonucleotides, and various rG4 concentrations) and a buffer with or without 10% of PEG8000 (Sigma, P5413). The effect of salt and PEG on LLPT was tested for 50 µM SERF2 mixed with equimolar TERRA12 or TERRA23. LLPT was measured in a non-crowding (20 mM NaPi, 100 mM KCl, pH 7.4) or crowding condition (20 mM NaPi, 100 mM KCl, pH 7.4 containing 2.5% or 5% w/v PEG8000) at varying G3BP1, HeLa cells extracted total RNA, SERF2, unstructured homo polyA RNA, structured HIV-1 TAR RNA and rG4 concentrations. For fluorescence imaging, 0.5% fluorescence-labeled (AF488, 6-FAM or Cy5) protein/rG4 samples were mixed into unlabeled sample mixtures. Sample imaging and FRAP measurements were done using a Nikon Ti2-E motorized, inverted microscope. This microscope is controlled by NIS Elements software and contains a SOLA 365 LED light source and a 100X oil immersion objective. Recovery half-life analysis was done using GraphPad Prism, and the image processing was done using Fiji ImageJ.

### Mutational analysis

5 µM of TERRA23 rG4s, dissolved in NaPi buffer, were mixed with equimolar concentrations of wild-type and different lysine-alanine mutant SERF2 and incubated for 30 min at room temperature. Gel-shift mobility assays were next performed by loading a 5 µM TERRA rG4 sample mixture, containing SERF2 and 20% glycerol, onto a 4−20% TBE gel (Invitrogen, EC6225). For the LLPT assay, 50 µM of TERRA rG4s was dissolved in 20 mM NaPi, 100 mM KCl, pH 7.4, containing 10% PEG8000 and mixed with different lysine-alanine SERF2 mutants at equimolar concentration. The samples were incubated for 30 min at room temperature, and DIC images were taken on a Nikon Ti2-E motorized, inverted microscope.

### Size-distribution analysis

*Size-exclusion chromatography*—20 µM of TERRA23 rG4s, dissolved in NaPi buffer, were mixed without or with 40 µM SERF2 and incubated for 30 min at room temperature. The mixture was then injected into a Superdex 200 Increase 10/300 GL size-exclusion chromatography column (Cytiva, 28-9909-44).

### Analytical ultra-centrifugation (AUC).

The AUC measurements were done for a SERF2 (9.4 µM) and TERRA12 rG4 (4.7 µM) sample mixture that was dissolved in 20 mM NaPi, 100 mM KCl (pH 7.4) at 22 °C. AUC measurements were done at 260 nm, where SERF2 has no absorption,

and with an intensity mode of 42,000 rpm. 420 µL samples were loaded into a two-channel epon-charcoal centerpiece, with a 1.2 cm path length, in an An60Ti rotor of a Beckman Optima Xl-I AUC. The data were analyzed with Ultrascan III software (version 4) and the LIMS server using the computing clusters available at the University of Texas Health Science Center and XSEDE sites at the Texas Advanced Computing Center.

## Structure calculation and all-atom MD simulations

SERF2 backbone assignments were done using standard 3D experiments, including HNCA, HNCACB, CBCAcoNH and HNCO experiments, along with the 2D HSQC collected at 4 °C. The side-chain assignment was carried out using a 3D $^{15}N$-resolved [$^{1}H$,$^{1}H$] TOCSY and a 2D TOCSY measured with a mixing time of 100 ms. The data were processed using NMRFAM-Sparky[118], and the dihedral angles were obtained utilizing the backbone carbon chemical shifts by the TALOS-N program[74]. Distance restraints for the structure calculation were obtained from 3D $^{15}N$-resolved [$^{1}H$,$^{1}H$] NOESY and a 2D NOESY measured with a mixing time of 200 ms and 150 ms, respectively. A total of 494 (intra−205, short−220 and medium−69) NOE distance constraints and 40 dihedral angles and 24 hydrogen bonds were used for the multiple-state ensemble calculation in CYANA 3.98.15. The NOEs are calibrated using the standard calibration in Cyana, where it uses the CH2 distances as 1.7 angstrom, and the rest is calibrated with respect to that. One hundred initial conformers were calculated using 10,000 torsion-angle dynamic steps and energy minimized. The 20 conformers, with the lowest target function values, were used to represent the calculated SERF2 structure. The 20 conformers were subjected to refinement using the Crystallography & NMR System or CNS constrained energy minimization in an explicit water model. The CYANA-generated SERF2 PDB coordinates, NOE constraints, dihedral angles, and hydrogen bond constraints were input to CNS for structure refinement. The 20 coordinates of the energy-minimized structures are deposited to the protein data bank as 9DT0, and the chemical shifts along with the restraints are deposited to the BMRB as 52141.

The MD simulations were performed using GPU nodes running in parallel. To gauge the intrinsic structural dynamics of these SERF2-TERRA rG4 systems, we performed all-atom MD simulations that were defined in a structure-based balanced forcefield, i.e., CHARMM36m[119] using GROMACS 2022.4[120]. The SERF2 structure was initially subjected to a 100 ns all-atom MD simulation in CHARMM36m force-field at pH 7.4 and 0.1 M KCl for structural refinement. All-atom MD simulations for 100 ns production run using CHARMM36m were initially done to test the structural stability of TERRA12 (PDB ID: 2KBP) and TERRA10 (PDB ID: 2M18) in solution containing 0.1 M KCl. The TERRA10 structure was next considered for protein-RNA complex simulation. The SERF2-TERRA rG4 complex structure was next built, using the HADDOCK program[79], by parsing the NMR distance restraints obtained from chemical shift perturbations and saturation transfer NMR data analysis. While side-chain chemical shifts or NOEs can further refine models, HADDOCK was explicitly designed to handle sparse experimental data, including backbone-only constraints, which have been shown to be sufficient for generating reliable models of protein-RNA and protein-protein complexes. The 15N-1H CSP-guided docking in principle can generate reasonable starting model[121], demonstrated in protein-protein or protein-RNA systems[108–111], that can further be subjected to long-timescale all-atom MD simulations, allowing to probe the complex's stability and interaction mechanism in atomic detail. A set of ambiguous active site SERF2 residues (T3, N5, R7, R11, Q12, K16, S19, S21, K23, A33, Q46, K47, A51, N52, K55 and E56) and TERRA rG4 guanines (G5, G9, and G10) obtained from chemical shift perturbation and saturation transfer difference NMR experiments were provided to the HADDOCK program to allow it to build the initial model structure of the complex. The energetically best cluster complex structure was next used for refinement and dynamic analysis. The complex structure was solvated in the TIP3P water model in an

orthorhombic water box. These model systems were electro-neutralized using 100 mM KCl, followed by energy minimization, using the steepest descent algorithms in less than 5000 steps, to remove steric clashes. The energy-minimized systems were then subjected to two-step equilibrations using NVT and NPT ensembles for 50 ns and keeping the temperature to 37 °C and pressure to 1 bar using a V-rescale thermostat and Berendsen barostat. These equilibrated systems were subjected to a final production MD of 0.5 µs. All the bonds involving hydrogen atoms were restrained using the SHAKE algorithm. The atomic coordinates of each system were saved every 100 ps, resulting in 5000 snapshots for post-dynamics analysis.

## MD simulation of SERF2-TERRA RNA G4 quadruplex liquid-liquid phase separation

We designed four densely packed, multi-component systems, containing 1:1 or 1:2 or 2:1 protein:RNA molecules, to study SERF2 and TERRA10 rG4 condensates using atomistic MD simulation. We designed two 1:1 MD systems, a large one comprising 30 protein and 30 rG4 molecules, and a small one that includes 6 protein and 6 rG4 molecules. The 1:2 and 2:1 system consists of 6 or 12 SERF2 or RNA molecules. The MD systems comprised randomly distributed multiple copies of SERF2, TERRA10, and 10% PEG8000 molecules solvated in a TIP3P water model in an orthorhombic water box with the dimensions of $20 \times 20 \times 20$ nm. To mechanize the complex molecular assembly and preparation of the input files for the MD simulation under periodic boundary conditions, we used a Multicomponent Assembler in CHARMM-GUI[122]. The small 1:1 system consisted of 6 SERF2, 6 TERRA, 60 units of PEG, 234 $K^+$ and 60 $Cl^-$, and 23185 water molecules. The 1:2 system has 6 SERF2, 12 TERRA10 rG4, 97 units of PEG, 506 $K^+$, 134 $CL^-$ and 35961 water; and the 2:1 system contains 12 SERF2, 6 TERRA10 rG4, 98 units of PEG, 230 $K^+$, 134 $CL^-$, and 36793 water molecules. The large system contained 30 units of SERF2, 30 TERRA10 rG4, 150 PEG, 1266 $K^+$ and 396 $Cl^-$, and 216499 water molecules. The input files for the MD simulations were generated using a CHARMM36m force field compatible with GROAMCS. The same MD simulation procedure involving energy minimization, two-step equilibration, and production was adopted, as described above, for both densely packed systems. The small system was subjected to 1 µs production MD, while the large systems were set to 0.5 µs. The trajectories of both systems were harvested at 100 ps intervals for post-dynamics analysis.

## Post-dynamic analysis of molecular dynamic trajectories

We employed the Gromos Clustering Algorithm method, with a cutoff of 0.2 nm, to obtain a representative conformation of the entire trajectory with maximum population. The hydrogen bond occupancy of each residue was computed using the hydrogen bonding analysis toolkit of Visual Molecular Dynamics (VMD 1.9.4a53). An angle cutoff of 30° and donor-acceptor distance of 3.5 Å was set to compute hydrogen bonds between protein and nucleic acids. The residue pairs (amino acid and nucleic acid) having higher percentages of hydrogen-bond occupancy were used for the generation of interaction plots. Structural representations were created using PyMOL and the BIOVIA Discovery Studio Visualizer. To probe the diffusion of SERF2 molecules in the small and large MD systems, the mean-square displacement of individual SERF2 molecules was calculated. The average end-to-end distance between the nitrogen atoms of T2 and A51 in SERF2 was computed using the 0.5 µs MD trajectory. The mean-square displacement analysis was carried out using gromacs *gmx msd* command. Cluster analysis, PEG molecule distribution within 6 Å from protein or RNA, time-resolved Hoogsteen H-bonds (donor atoms: guanine N1, N2 and O2; acceptor atoms: N7 and O6) formation and contact map analysis in the large MD system comprising 30 SERF2 and 30 TERRA10 rG4 was performed using MDanalysis program[123].

## Single-molecule fluorescence microscopy and droplet fusion by optical tweezers

Single particle tracking (SPT) and fluorescence resonance energy transfer (FRET) experiments were conducted using a double cysteine SERF2 mutant (A2C and A51C) labeled with Cy3 and Cy5 fluorophores. 200 μM of double-mutant SERF2 (A2C, A51C) was fluorescently labeled by incubating it with 5x molar excess of Cy3 and Cy5, applying the same protocol described above in the section 'Protein labeling'. The occupancy of the Cy3 and Cy5 in each SERF2 molecule was not determined because the SPT and FRET measurements do not rely on a uniform 1:1 Cy3:Cy5 labeling for studying SERF2-TERRA12 rG4 condensates. For single-molecule experiments, droplets were prepared by mixing 100 μM unlabeled SERF2 and TERRA12 rG4 containing a picomolar concentration of Cy3-Cy5 SERF2 or diluted as needed to obtain a sparse concentration where single molecules are distinguishable for SPT. The sample chamber was prepared by mounting #1.5 coverslips on a standard glass microscope, using double-sided tape placed about 5 mm apart as spacers. The chamber was coated with ~15 μL of 0.1 mg BSA (5 min) followed by rinsing thrice with NaPi buffer containing 10% PEG8000. SERF2-TERRA12 rG4 droplets were then injected into the sample chamber, and droplets were allowed to settle for ~20 min. An $O_2$-scavenging system (1 mM trolox, 2.5 mM protocatechuic acid (PCA), and 7 μg/mL protocatechuate-3,4-dioxygenase (PCD)) prepared in NaPi, PEG8000 buffer was then applied to the sample chamber immediately before performing the SPT and FRET measurements at room temperature. Fluorescent images were acquired with total internal reflectance fluorescence (TIRF) illumination (using an Olympus CellTIRF module). Acceptor and donor fluorescent images were collected simultaneously on dual Andor Ultra EMCCD cameras at frame rates between 10 Hz and 100 Hz. The data were analyzed using an in-house MATLAB script to obtain the diffusion constants via a linear fit of the mean squared displacement for each particle and FRET efficiency.

The fusion of SERF2 and TERRA rG4 droplets was studied using a bespoke dual-trap optical tweezers instrument equipped with a bright-field camera and a 60X oil immersion objective (Nikon APO TIRF 60X). Two optical traps are generated by splitting a 10 W 1064 nm laser by polarity, with each polarized beam steered by an acousto-optic deflector (IntraAction DTD-274HD6). Each particle was trapped using approximately 250 mW laser power. 100 μM of SERF2 was mixed with equimolar TERRA23 or TERRA12 rG4s in 20 mM NaPi, 100 mM KCl, pH 7.4 buffer containing 10% PEG8000 (w/v). The glass slide chamber was prepared as described above and coated with 0.1 mg BSA (5 min), followed by rinsing. 15 μL of SERF2-rG4 droplet samples were passed into a glass slide chamber. Particle fusion was observed by trapping two condensate droplets in separate optical traps and steering the laser foci together until particle fusion was observed. Movies of fusion events were recorded within ~5 min at room temperature to avoid droplet settlement on the glass slide and droplet aging. The droplet fusion time was extracted using the ImageJ program.

### Reporting summary

Further information on research design is available in the Nature Portfolio Reporting Summary linked to this article.

## Data availability

The atomic coordinate generated in this study has been deposited in the Protein Data Bank (PDB) under the accession code 9DT0 (Human SERF2). The source data underlying Figs. 1–8, and Supplementary Figs. 1–7, 10, 11, 13, and 14 are provided as a Source Data file. Source data are provided with this paper.

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

## Acknowledgements

The authors acknowledge the CERIC-ERIC consortium for access to experimental facilities and financial support. NMR data reported in this study were supported by the University of Michigan BioNMR Core Facility. We thank Dr Anthony Vecchiarelli for providing the instrumentation facility with fluorescence imaging. We thank the National Brain Research Centre (NBRC), India, for providing a High-Performance Research Computing facility. This research was supported by the Howard Hughes Medical Institute (HHMI) to J.C.A.B., start-up funds from the University of North Carolina and the National Institutes of Health (R35GM142864) to D.D., JSPS KAKENHI (20H05626) to H.S., and the Slovenian Research and Innovation Agency (ARIS, grants P1-0242 and J1-60019) to V.K. and J.P.

## Author contributions

B.R.S. and J.C.A.B. conceived the original idea; J.C.A.B. supervised the research; B.R.S., X.D., E.W., N.W.C., B.B.Z., and E.M. performed the experiments in this study. B.R.S., X.D., E.J., E.M., B.B.G., H.Y., V.S., V.K., J.P., S.E.H., B.D., A.K., J.D.H., H.S., and D.D. performed data analysis. B.R.S., X.D., and J.C.A.B. wrote the manuscript. The manuscript was written through the contributions of all authors. All authors have approved the final version of the manuscript.

## Competing interests

The authors declare no competing interests.
