## [Peer Review file · Nature Communications]

Visualization of liquid-liquid phase transitions using a tiny G-quadruplex binding protein

Corresponding Author: Professor James Bardwell

Version 0:

Reviewer comments:

Reviewer #1

(Remarks to the Author)

The authors study the interactions of the stress granule protein SERF2 with G quadruplex RNA (rG4) in solution and the role of SERF2 in liquid-liquid phase separation by combining biochemical, biophysical and NMR structural analysis, microscopy and cellular data. The topic is highly interesting and relevant and choosing a seemingly well-behaved systems of small protein and RNA components to explore their structure and interactions by NMR is nice. The authors provide a wealth of experimental data but the quality and how conclusions are drawn from these is not always clear, and sometimes confusing. A number of statements are not well supported by the data shown. This starts with the title, which is too generic, but also the suggestion that high resolution structural information is provided that report on RNPs in biomolecular condensates, which I am not convinced of. The authors mainly study interactions in the soluble form and then infer their relevance for the condensate by mutational analysis. The title and abstract raise the expectation that experimental data are provided on the RNP in the condensate phase, for which only limited information is presented, e.g. from single particle tracking and otherwise from MD simulations, which are ambitious, challenging and difficult to judge (for this reviewer). In my view, the manuscript presents interesting data on a relevant and important system, but the presentation and documentation of the data is often not complete and not clear. A number of conclusions are not well supported by the data. The authors should improve the presentation but also provide additional experimental evidence for some of the key conclusions.

Specific comments:

- The title is very generic to the extent of being misleading. The title should be adjusted to include information on the specific systems (SERF, G4) studied
- Line 92: Binding affinities of SERF2 with rG4 seems rather low (micromolar) and strongly salt dependent, this should be commented on (Fig. S1)
- Fig. 2 the comparison of phase separation of SERF2 with total HeLa cell RNA vs specific rG4 is puzzling. The microscopy seems to indicate aggregation with total cell RNA (the authors state gel-like, why?) - while nice droplets are form with individual rG4? Th authors should comment on this.
- The authors propose that FRAP recovery is faster with specific rG4 compared to cellular RNA which comprises other RNA structures. As a control to provide some support for this interpretation, both binding affinities (FP, EMSA) and FRAP recovery should be done with some structured RNAs (that are not G4).
- The authors claim a "high-resolution" structure of SERF2, However, what they present is an ensemble of a helical region with flanking disordered IDRs. It is unclear how this ensemble was generated, e.g 494 NOEs were used, these are presumably largely sequential intrachain NOEs, what is more interesting are those that my define the helical region, these should be shown and discussed. How were NOEs calibrated?
- A structural statistics table is missing.
- NMR R2 and R1 and secondary chemical shift data should be shown with and w/o RNA, it is unclear how the helical region is supported by the data. The authors should also measure {1H}-15N heteronuclear NOE data to highlight rigidity of the helical region.
- The chemical shift perturbations upon TERRA rG4 binding are very small. The authors should show spectral zooms to illustrate the spectral changes.
- The authors should also monitor 1H,13C correlations for protein signals to map the interface with RNA and potentially obtain intermolecular NOE contacts.
- Fig. S4. The saturation transfer experiments are interesting, but more details should be provided about the experimental setup. Moverover, can the authors exclude that transfer it mediated by solvent exchange? This should be clarified.

- Fig. S4h and explanations are unclear. The authors should indicate amino signals of the RNA.
 - Fig. S4j: it seems that TERRA alone is also highly heterogeneous. Which species is then observed by the NMR imino spectra, which seem to indicate a single conformation?
 - The utility and significance of the HADDOCK modelling is unclear to this reviewer. There are only fuzzy restraints included, i.e. mapping contacts of N-terminal IDR residues with the RNA nucleotides. That these regions are seen to interact in the structural ensemble is expected and does not prove the validity. More importantly, the authors mention several hydrogen bonds of basic residues to rG4 nucleotides. How are these experimentally supported? The authors should analyze chemical shift changes of the side chains involved, changes in solvent exchange or, in the best case, identify intermolecular NOEs to support these interactions.
 - Fig. 3: the structural representation is difficult to read, the contacts described are barely visible. The presentation of the structural model should be improved.
 - The authors provide support for the RNA contacts involving positively charged side chains based on mutational analysis studying RNA binding by NMR and gel shift assays. They then show that combinations of these mutations impart phase separation in vitro to conclude that this involves multivalency interactions, as being an important mechanism for phase separation. This is overstated for multiple reasons: 1) the contacts are studied in the soluble complex, no evidence is provided in a condensed phase, 2) that multiple contacts are important may just reflect overall higher RNA binding affinity and not necessarily indicate that multivalency is important for phase separation!?
 - Have the authors attempted to use NMR to study a dense phase sample or a mixed phase sample to obtain insights into differences in relaxation, NOE contacts or diffusion rates?
 - Fig. 4: it is unclear which (experimental) data support the model shown and discussed. I cannot judge the validity of the MD simulations, however, in any case some support for the conclusions based on these simulations will be important. For example, the authors should identify different interfaces relevant to form the high order complexes and explore their contributions by mutational analysis where consistently binding affinity in solution and effects on phase separation in vitro, and if possible, in cells should be monitored and correlated
 - The cellular data exploring the role of SERF2 for stress granules are merely phenotyping. How can these be correlated with the in vitro data?
- Line 417-419: the authors state that SERF2 may enhance the fluidity of stress granules through LLPS. I do not see why this should be and how this is supported by any of the data.
- Fig. 7: co-condensation should be shown by using different dyes. It is unclear what the authors mean by SERF2 facilitating phase transitions of G3BP1. What is shown are effects on FRAP recovery. How this relates to fluidity changes and especially how the molecular and structural features discussed before are relevant here is unclear.

Reviewer #2

(Remarks to the Author)

Summary:

This study explores the molecular and structural basis of stress granule formation through high resolution analyses of the small human protein SERF2 and its specific interactions with RNA G quadruplexes (rG4s). The authors employ a combination of NMR spectroscopy, fluorescence assays, single-molecule imaging, and all-atom molecular dynamics (MD) simulations to show how SERF2 engages rG4s and participates in liquid-liquid phase separation. They demonstrate that SERF2's disordered regions mediate multivalent interactions with rG4s, which are important to drive LLPT under physiologically relevant conditions. They further show that SERF2 promotes the formation and fluidity of stress granules in vivo. This work addresses an important and timely question at the interface of structural biology, RNA biophysics, and cellular phase transitions. The authors make a significant stride in this direction. However, several key clarifications and additions are needed before the manuscript can be accepted.

Major comments:

1. Why does the system form a hydrated core? This is not addressed completely in the manuscript. The formation of such a ring-like structure seems counterintuitive to having the two components mixed. Is there any experimental evidence to support this structure or is it only seen in simulation?
2. How do the authors make sure 0.5 us is sufficient time for the simulation? Is this ring-like structure a metastable structure that would disappear after a longer simulation?
3. What is the role of PEG? Are there any PEG-protein/RNA interactions in the system? Is all the PEG outside the droplet or is it present within the hydrophilic core?
4. Why do the authors not use a backmapping strategy to ensure that the system is pre equilibrated before starting simulations at atomistic resolution? Simulating the system using MARTINI could allow for the system to equilibrate faster which could then be used to backmap and study atomic interactions.
5. How dynamic is the ring-like structure? Do SERF/RNA molecules within the structure exchange with the surroundings or not? If not, how do the authors conclude that these are liquid-like assemblies?
6. The authors observe striking ring-like assemblies in MD simulations involving multiple SERF2 and rG4 molecules. These

structures are interpreted as representative of early phase condensates. Are there any microscopy data that might support this structural formation?

7. The MSD vs. lag time plots for the atomistic condensed phase should be shown. Do these plots show an exponent of 1 indicative of diffusive motion or is the motion of SERF2 arrested in their simulations?

8. Within the condensed phase, do the RNA chains maintain their quadruplex structure? In the dilute phase, the authors observe less stability than experiments. Does the condensed phase stabilize the structure more or favor unfolding?

Minor comments:

The section regarding whether crowders are used in the MD simulation in the condensed phase is unclear. Going through the methods, it is clear that crowders were added, but the text in the results section is very confusing. The authors could simply state that they add the crowder to match the NMR conditions.

Reviewer #3

(Remarks to the Author)

This study examined the structural details of the protein SERF2 and its interactions with RNA G-quadruplexes, in the context of phase separation. The authors first showed that SERF2 can specifically bind to rG4 structures. Using high-throughput sequencing methods, they showed that the SERF2 prefer to interact G-rich RNAs that are likely to form stable rG4 structures, including several known rG4 sequences. In vitro, SERF2 can form liquid droplets with G4 or HeLa cell total RNAs. However, SERF2-rG4 droplets are less dynamic, both protein and RNA exchange slower buffer compared to SERF2-total RNA droplets.

Using NMR chemical shifts and NOE restraints, the authors defined the structure of SERF2 in solution, which consists of a helical region flared by disordered N- and C-term regions. Protein CSP analysis and relaxation measurements revealed that both the N- and C-term disordered regions mediate SERF2's interaction with the model rG4 TERRA, where the N-terminus drives the binding, and the C-terminus coordinate the complex formation. Size-exclusion chromatography showed that rG4 can form 1:1 and 1:2 multimeric complexes with SERF2.

Using docking program, the authors proposed model structures for 1:1 and 1:2 TERRA-SERF2 complex and subsequently refined them using all-atom simulation. Consistent with the NMR results, the complex structure is mediated by the N-term disordered residues in SERF2. The complex structures revealed details such as H-bonds between R and K residues in SERF2 and bottom guanines in TERRA, additional amino acids are involved in binding in the 1:2 complex, and that protein-RNA bonding can disrupt G-G pairs to destabilize rG4 structure. The destabilization of rG4 structure upon binding to protein is further verified by CD, though the extent was overestimated in the MD simulation. To verify amino acids involved in SERF2-rG4 binding, the authors carried out mutational study and confirmed the importance of these key residues identified in NMR and MD studies.

Next, to study the RNA-protein co-phase separation, the authors simulated RNA-protein condensate using all-atom simulation. In addition to unbound rG4 and SERF2-TERRA rG4 oligomers, a ring-like higher order structure was identified. FRET data suggested that SERF2 adopts a more compact structure within the condensate, consistent with MD simulation results.

In multiple cell lines, the authors showed that depletion of SERF2 drastically reduce the number of induced stress granules. The extent of impact is comparable to knockdown of G3BP1+G3BP2. Additionally, cellular granules formed under SERF2 depleted conditions are more like aggregates than condensates.

Comments:

Overall, I think this manuscript presents a well-rounded study of the SERF2-rG4 interactions in both solution and phase separation. The experimental and simulation studies were designed to complement each other and combine to produce an in-detail structural picture of the complex in various conditions. I found the interpretation of data to be reasonable. However, I do have a few concerns, see below:

- The authors need to specify the TERRA RNA sequence used in each assay and provide justification for not using the same TERRA sequence. I.e., in the 1D ¹H NMR titration, the authors apparently used TERRA12, whereas in the HADDOCK/All-atom modeling, the authors switched to a shorter TERRA10 sequence. This is problematic because various TERRA sequences can form different structures: TERRA12 forms a dimeric, 3-layer G4 structure while TERRA10 forms a tetrameric, 5-layer structure. Because of the distinct structures, the SERF2-rG4 binding modes might not be the same. Had the author used the same sequence in these studies, we would have been able to compare amino acid/RNA bases that participate in the binding to see if they are consistent.
- I wonder if the ring-like tertiary structure seen in the condensed phase simulation is a result of the specific protein and RNA numbers and their ratio in this set up, or a general phenomenon. The authors should attempt the same experiment with different protein/RNA counts to test this.

Minor comments:

- Ca, Cb secondary shifts should be reported from the SERF2 assignments, in order to provide further evidence for secondary structures in the protein.
- PEG molecules are shown in figure 4a only at 0 s but not 0.5 s. It would be interesting to see how the PEG molecule exists in the simulated condensed phase and whether they interact with the protein.
- In Figure 4c the two 2:1 complex structure in the simulated condensed phase and the dilute phase structures are oriented in a way that is difficult to see. Slight adjustment should help to draw a better comparison between them.

- The data in figure 7e does not appear to support the claim that SERF2 enhances G3BP1 fluidity. While in the microscopy image the droplet with SERF2 does appear to recover faster, statistical comparison in 7e suggest that the difference is within error.
- “condensation is mainly due to an increase in the valency of interactions driven by crowding” – I think the term valency refers to the potential number of sites for interaction. So, crowding does not change that (it does not change the (primary) structure). So, I would rephrase this.

Reviewer #4

(Remarks to the Author)

Reviewer #5

(Remarks to the Author)

Version 1:

Reviewer comments:

Reviewer #2

(Remarks to the Author)

The authors have addressed all of my comments and concerns satisfactorily and I recommend this article for publication in Nature Communications.

Reviewer #3

(Remarks to the Author)

I appreciate the authors' thorough revision. Most of my previous concerns have been effectively addressed. I particularly appreciate the inclusion of additional data comparing SERF2 interactions with TERRA RNA of different lengths, and the MD simulation across various protein:RNA stoichiometries.

That said, I strongly suggest that the authors further address a few minor points in the presentation for clarity.

Minor:

In Figure S5, some of the green bar for SERF2-TERRA CSPs are covered by the blue bars, re-plotting so both are visible would be helpful.

The new Figure S6 would benefit from including also a set of peak intensity plots, which would help to show if the same residues are involved in binding TERRA RNAs of different lengths.

Reviewer #4

(Remarks to the Author)

Reviewer #5

(Remarks to the Author)

We sincerely thank the reviewers for their thorough evaluation of our manuscript and for the constructive and insightful comments provided. The feedback has been invaluable in strengthening both the clarity and scientific rigor of our work. In response, we have revised the manuscript substantially and addressed all comments point by point below.

We would first like to highlight experimental additions:

1. We performed phase separation assays for G3BP1 and SERF2 using different fluorophores to demonstrate their colocalization. We also performed SERF2 phase separation and FRAP assays in the presence of RNA and unstructured and structured non-G4 forming RNA sequences. We now show SERF2 binding assays for unstructured non rG4 forming RNA sequences (CCCCGG and mutated TERRA “UUACCG”) and different lengths of TERRA sequences.
2. We performed 1D titration, 15N-1H, 13C-1H HSQC, and saturation transfer NMR experiments on different TERRA length rG4s to validate protein-RNA interactions.
3. We did MD simulations at various protein-RNA stoichiometries and performed quantitative MD analysis including the role of crowders and stability of rG4s in the large simulation setup.
4. We conducted CD spectroscopy to monitor changes in SERF2’s structure upon binding to RNA.

A complete point by point response follows. We hope these revisions meet the expectations of the reviewers.

Reviewer #1 (Remarks to the Author)

The authors study the interactions of the stress granule protein SERF2 with G quadruplex RNA (rG4) in solution and the role of SERF2 in liquid-liquid phase separation by combining biochemical, biophysical and NMR structural analysis, microscopy and cellular data. The topic is highly interesting and relevant and choosing a seemingly well-behaved systems of small protein and RNA components to explore their structure and interactions by NMR is nice. The authors provide a wealth of experimental data.

Response: We sincerely thank Reviewer #1 for their thoughtful and constructive feedback. We appreciate the recognition of the relevance and interest of our study.

The quality and how conclusions are drawn from these is not always clear, and sometimes confusing. A number of statements are not well supported by the data shown. This starts with the title, which is too generic, but also the suggestion that high resolution structural information is provided that report on RNPs in biomolecular condensates, which I am not convinced of. The authors mainly study interactions in the soluble form and then infer their relevance for the condensate by mutational analysis. The title and abstract raise the expectation that experimental data are provided on the RNP in the condensate phase, for which only limited information is presented, e.g. from single particle tracking and otherwise from MD simulations, which are ambitious, challenging, and difficult to judge (for this reviewer).

In my view, the manuscript presents interesting data on a relevant and important system, but the presentation and documentation of the data is often not complete and not clear. A number of conclusions are not well supported by the data. The authors should improve the presentation but also provide additional experimental evidence for some of the key conclusions.

Response # We also appreciate the reviewer's comments highlighting areas that require clarification, improved presentation, and stronger experimental support. We have carefully revised the manuscript to address these concerns, including altering our title and abstract to better reflect the scope of our data, improving the clarity of data presentation, and providing additional experimental evidence where necessary. Below, we provide a detailed point-by-point response to each comment.

Specific comments:

1. *The title is very generic to the extent of being misleading. The title should be adjusted to include information on the specific systems (SERF, G4) studied.*

Response #1 We thank the reviewer for this helpful suggestion. We agree that the original title may be too general. We have revised the title to: "Visualization of liquid-liquid phase transitions using a tiny G-quadruplex binding protein". We choose not to use the word "SERF" in the title as titles that include unfamiliar names or abbreviations are unclear to the general reader.

2. *Line 92: Binding affinities of SERF2 with rG4 seems rather low (micromolar) and strongly salt dependent, this should be commented on (Fig. S1).*

Response #2 Overall SERF2-rG4 interactions are in the low micromolar range and show salt dependence, as expected for RNA interactions with highly charged, disordered proteins. We do note that SERF2 binds sub-micromolar affinity to TERRA rG4 repeats and included the suggested statement regarding salt dependence in the revised Figure S1 caption.

3. *Fig. 2 the comparison of phase separation of SERF2 with total HeLa cell RNA vs specific rG4 is puzzling. The microscopy seems to indicate aggregation with total cell RNA (the authors state gel-like, why?) - while nice droplets are form with individual rG4? The authors should comment on this.*

Response #3 We thank the reviewer for asking us to comment and clarify the distinct morphologies found for SERF2 mixed with total RNA and rG4. We have now characterized the dynamic and reversible nature of SERF2 and RNA condensates using FRAP. SERF2 bound rG4 or total RNA condensates show FRAP recovery times within a few minutes. This reveals that SERF2-total RNA structures are not aggregates. We agree with the reviewer that the word gel-like is not specifically supported by our observations. We have therefore now termed them irregular shaped condensates to more accurately reflect our experimental observations (Line 142-145). We also have now noted in the manuscript that similar irregular condensates that have similar FRAP recovery times have been previously observed for proteins like FUS and G3BP1 (Weirui et al., 2021, eLife 10:e64252 Guillén-Boixet et al., 2020, Cell, 181, 346-361).

4. *The authors propose that FRAP recovery is faster with specific rG4 compared to cellular RNA which comprises other RNA structures. As a control to provide some support for this interpretation, both binding affinities (FP, EMSA) and FRAP recovery should be done with some structured RNAs (that are not G4).*

Response #4 We didn't mean to imply that FRAP recovery is faster with rG4 sequences than other structures, rather we previously showed the opposite result. To further examine this phenomenon we have now, as suggested, studied the dynamic nature of SERF2 droplets formed upon binding additional structured and unstructured non-rG4 sequences (Line 168-175). In Figure S3e, we have observed a faster recovery of SERF2 mixed with random length polyA sequences, and a relatively slow recovery for HIV-1 TAR structure RNA. We have included new results in Figure S3e and h. Additionally as commented, we

also have tested SERF2 binding to non-rG4 sequences that include a 4-repeat CCCCCG and mutated 4-repeat TERRA (UUACCG) and structured HIV-1 TAR RNA (Line 104-108) all of which show undetectable binding using fluorescence polarization assay (Fig. S2c).

5. *The authors claim a "high-resolution" structure of SERF2. However, what they present is an ensemble of a helical region with flanking disordered IDRs. It is unclear how this ensemble was generated, e.g. 494 NOEs were used, these are presumably largely sequential intrachain NOEs, what is more interesting are those that may define the helical region, these should be shown and discussed. How were NOEs calibrated?*

Response #5 We appreciate the reviewer's thoughtful comments regarding our use of the term "high-resolution" and the presentation of the SERF2 structural ensemble. We agree that this warrants clarification.

In our study, we generated an ensemble structure of SERF2 based on solution NMR data, which revealed a central α -helical region flanked by intrinsically disordered regions. While we initially described this as a "high-resolution" structure, we recognize that this terminology may imply a level of atomic precision more appropriate for fully folded globular domains. To avoid overstatement, we have revised the manuscript to more accurately describe this as an "NMR-derived ensemble model" or "structural ensemble of the helical core and disordered flanks," or describe our ensembles as "detailed" more consistent with the IDP nature of SERF2.

For our calculations, the NOE's are calibrated using the standard calibration in Cyana where it uses the CH₂ distances as 1.7 angstrom (script obtained from https://nesqwiki.chem.buffalo.edu/index.php/NOE_Calibration_Using_CYANA#NOE_Calibration_Using_CYANA_2.1) and the rest is calibrated with respect to that. We have now included these details in the methods (Line 1003-1005) and provided a statistical table of the restraints used for structure calculations (Table 1) We also show the NOE SeqPlot in Figure S4c.

6. *A structural statistics table is missing.*

Response #6 A structural statistics table (Table 1) has now been included to the revised manuscript.

7. *NMR R₂ and R₁ and secondary chemical shift data should be shown with and w/o RNA, it is unclear how the helical region is supported by the data. The authors should also measure {¹H}-¹⁵N heteronuclear NOE data to highlight rigidity of the helical region.*

Response #7 We thank the reviewer for these valuable suggestions. We have now included per-residue R₂, R₁ and ¹⁵N-¹H heteronuclear NOE relaxation data (Line 268-272) for SERF2 in both the free and RNA-bound states (Figure S7(b-d)). We have further tested for changes in protein secondary structure occurring in the presence of rG4 by CD spectroscopy, but observed no substantial change in the protein helicity (Line 282-284; Figure S7f). This is consistent with this, {¹H}-¹⁵N hetNOE values in the helix region remain moderately high (~0.3-0.5) in both free and bound states, and R₁ values are largely unchanged, indicating preserved helix rigidity. The observed increase in R₂ upon RNA binding including the helix region (residue 33-47) suggests local conformational exchange or slow dynamics in the intermediate regime that occur upon RNA binding, rather than changes in backbone order or secondary structure.

8. *The chemical shift perturbations upon TERRA rG4 binding are very small. The authors should show spectral zooms to illustrate the spectral changes.*

Response #8 We thank the reviewer for this suggestion. We have now included zoomed-in regions of the spectra (Figure 3b) to clearly illustrate the observed shifts of several residues as suggested. Although the chemical shift perturbations upon TERRA rG4 binding are modest at low stoichiometry, we observe strong perturbation and signal broadening at high rG4 concentration, as expected for interactions involving intrinsically disordered proteins.

9. The authors should also monitor ^1H , ^{13}C correlations for protein signals to map the interface with RNA and potentially obtain intermolecular NOE contacts.

Response #9 We thank the reviewer for this valuable suggestion. Following this recommendation, we have now run ^{13}C HSQC experiments for the SERF2 protein in the absence or presence of rG4, and the chemical shift changes we observed are minimal (please see Image-1, right panel). We also tested if SERF2's conformation is altered upon rG4 binding. Our CD spectroscopy results showed a very little structural change (please see Image-1, left panel) which resembles to our previous finding for the SERF homologous

SERF2+TERRA rG4

SERF2

Image 2: 3D NOESY-HSQC (H-1/N-15 correlation via double inept transfer with N-15 (and C-13) filter) spectra of SERF2 alone on bottom row and SERF2+TERRA rG4 (top row).

protein human Znf706 upon its binding to DNA G-quadruplexes (Nucleic Acids Res. 2024, 52, 4702-4722). We did not include these results in the manuscript, but could if requested.

We attempted to acquire intermolecular NOEs using Bruker program NOESYHSQCF3GPWG13D (H-1/N-15 correlation via double inept transfer with N-15 (and C-13) filter) and have no success in obtaining intermolecular NOEs in the complex state. We find this is challenging for this protein-RNA system (please see Image-2), and making the determination of a high-resolution structure derived from NOE constraints and correctly assigning them (which require specific labeling strategy) is very challenging for us to pursue.

10. *Fig. S4. The saturation transfer experiments are interesting, but more details should be provided about the experimental setup. Moreover, can the authors exclude that transfer is mediated by solvent exchange? This should be clarified.*

Response #10 We thank the reviewer for making this important point. We have now included additional details, as suggested in the methods (Line 903-914). To address the concern regarding potential saturation transfer occurring via solvent exchange, we have compared the magnetization observed in the SERF2-RNA complex with RNA alone or protein alone (Line 299-316). We choose to saturate the regions (imino peaks in RNA or A33 residue amino peak in protein) and see very weak or no intermolecule magnetization transfer suggesting the exchange observed in the protein-RNA complex are not exchanging or being saturated by water directly. These control experimental results are now shown in Figure S9, and experimental details are described in the revised methods section.

11. *Fig. S4h and explanations are unclear. The authors should indicate amino signals of the RNA.*

Response #11 We thank the reviewer for pointing this out. Our intention was to note that the SERF2 A33 amide proton (~8.5 ppm) does not overlap with RNA amino or imino protons. However, we agree the reference to imino protons was potentially misleading given their appearance in the downfield part of the spectrum. We have revised the figure caption to remove mention of imino protons and revised the explanation for clarity (Figure S4 is now updated as Figure S9).

12. *Fig. S4j: it seems that TERRA alone is also highly heterogenous. Which species is then observed by the NMR imino spectra, which seem to indicate a single conformation?*

Response #12 We regret not clarifying this well enough in the figure captions. We have used different length of TERRA RNA sequences for our study. The reason for this is that the short TERRA12/TERRA10 RNA is structurally resolved, forms dimeric/tetrameric rG4 structures, and shows well-resolved imino protons suitable for structural studies (Figure S8). On the contrary, the long single-strand TERRA sequences that from intramolecular rG4 folding is biologically more relevant (Biffi et al., 2012, J Am Chem Soc. 134(29):11974-11976), but are relatively heterogeneous with poorly resolved imino signals as observed for TERRA23. We have now explained this in the main-text and included the choice of using different TERRA RNA lengths to address specific questions in the revised manuscript (Line 225-240). We are also careful to specify the TERRA length whenever appropriate.

13. *The utility and significance of the HADDOCK modelling is unclear to this reviewer. There are only fuzzy restraints included, i.e. mapping contacts of N-terminal IDR residues with the RNA nucleotides. That these regions are seen to interact in the structural ensemble is expected and does not prove the validity. More importantly, the authors mention several hydrogen bonds of basic residues to rG4 nucleotides. How are these experimentally supported? The authors should analyze chemical shift changes of the side chains involved, changes in solvent exchange or, in the best case, identify intermolecular NOEs to support these interactions.*

Response #13 We appreciate the reviewer's comments regarding the HADDOCK modeling. This model was guided by experimentally derived restraints from CSPs and saturation transfer experiments. While intermolecular NOEs, if obtainable, could provide strong structural validation, we found them very difficult to obtain for our system as is commonly observed in heterogeneous complexes, especially when involving intrinsically

disordered regions or RNA elements prone to flexibility and conformational averaging. As we provided details in the previous response, ¹⁵N-edited ¹³C-filtered NOESY-HSQC failed to provide us with useful information. While side-chain-specific chemical shift assignments or NOEs would provide further resolution, they are technically challenging for large protein-RNA complexes involving disordered regions. We note that, even the calculation of NMR structures based on NOEs for small ligands binding to G4s is challenging due to NMR exchange regime and low residence time of ligands on G4. This has been discussed in literature (*Bessi et al., 2017, Targeting G-quadruplex with small molecules: An NMR view. Modern Magnetic Resonance. Springer, Cham. https://doi.org/10.1007/978-3-319-28275-6_126-1*). Disordered proteins further complicate the issue, their spectrum complexity and lack of stable tertiary structure pose significant hurdles for NOE assignment, which relies on well-defined inter-proton distances.

To overcome these limitations, we employed molecular dynamics (MD) simulations to refine the initial HADDOCK models. MD simulations enable structural relaxation of complexes and sampling of atomic-level interactions, such as hydrogen bonding patterns between basic side chains and rG4 guanines. These patterns were consistently observed across the trajectory, suggesting stable interactions that align with the electrostatic and chemical nature of the binding interface. Our CSP data, combined with MD-refined models, represent a mechanistically plausible and biophysically consistent framework for protein-rG4 recognition. We have also now experimentally tested the functional significance of the basic residues that the MD-models indicated to be interacting with RNA using combinatorial mutagenesis and binding assays, which showed reduced RNA binding affinity, supporting their involvement in binding.

We have now clarified these points in the revised manuscript (Methods and Discussion sections, Line 1025-1031; Line 640-646) and added a note acknowledging the limitations and potential future directions involving more extensive NMR-based validation.

14. *Fig. 3: the structural representation is difficult to read, the contacts described are barely visible. The presentation of the structural model should be improved.*

Response #14 We have revised Fig. 3 to enhance clarity by indicating key contact residues by arrows in the complex structure models.

15. *The authors provide support for the RNA contacts involving positively charged side chains based on mutational analysis studying RNA binding by NMR and gel shift assays. They then show that combinations of these mutations impart phase separation in vitro to conclude that this involves multivalency interactions, as being an important mechanism for phase separation. This is overstated for multiple reasons: 1) the contacts are studied in the soluble complex, no evidence is provided in a condensed phase, 2) that multiple contacts are important may just reflect overall higher RNA binding affinity and not necessarily indicate that multivalency is important for phase separation!?*

Response #15 We appreciate the reviewer's thoughtful critique. We agree that RNA-protein interactions in the dilute phase may not directly reflect behavior in the condensed phase as our data are primarily guided by simulation findings. Due to the inherent complexity and heterogeneity of condensates, obtaining precise binding affinities under crowding or phase-separating conditions remains technically challenging and may yield misleading binding affinity values. Our conclusion regarding multivalency is based on the observation that combinatorial mutations disrupting multiple basic patches as predicted from the all-atom MD models significantly impair both RNA binding and phase separation, suggesting that multivalent interactions are functionally coupled. Nonetheless, we have revised the text to acknowledge these limitations and now frame our interpretation more cautiously, emphasizing that the mutations reduce RNA-driven condensation likely by weakening

cumulative electrostatic interactions, rather than proving multivalency per se as the sole mechanism.

16. *Have the authors attempted to use NMR to study a dense phase sample or a mixed phase sample to obtain insights into differences in relaxation, NOE contacts or diffusion rates?*

Response #16 No, we have not performed NMR measurements on dense or mixed phase samples due to significant line broadening and signal loss under those conditions, which greatly limits structural insights. At some point in the future, advanced NMR and labeling strategies optimized for LLPS systems, may allow for probing relaxation properties, diffusion, or intermolecular NOEs in the condensed phase which would provide valuable information but unfortunately, we are not there yet.

17. *Fig. 4: it is unclear which (experimental) data support the model shown and discussed. I cannot judge the validity of the MD simulations, however, in any case some support for the conclusions based on these simulations will be important. For example, the authors should identify different interfaces relevant to form the high order complexes and explore their contributions by mutational analysis where consistently binding affinity in solution and effects on phase separation in vitro, and if possible, in cells should be monitored and correlated.*

Response #17 We appreciate the reviewer's feedback and agree that experimental validation of computational models is important. We found that the diffusion of proteins within droplets in our single molecule experiments and our MD simulation studies are very nicely correlated helping to validate our results. This approach was also used to validate similar MD experiments in recent literature (Galvanetto et al., Nature, 2023, 619, 876-883; Rekhi et al., Nature Chemistry, 2024, 1113-1124). The SERF2 and rG4 binding interfaces that we found from simulations and experiments in dilute solution appear to be conserved to the condensed phase (Figure 4b and 4c). As illustrated in Figures 3i and S10f, when we target multiple lysine residues (K16, K17, K23-K25) in SERF2 that are evolutionary conserved in SERF family proteins we observe a weak binding in solution state, and they also effect liquid-liquid phase separation in crowding state. For instance, single mutations at the conserved sites K23 or K25 have a little impact, while making multiple mutations does, implying that multiple contacts are involved. While extending this combination of mutations could provide more detailed insights, we think we have been able to show that multiple contacts are likely involve in phase separation, and have already been able to study seven different regions. C-terminal contacts Q46-K47 for instance have minimal effects, while N-terminal contacts show stronger effects both on binding in solution and phase separation in crowding conditions. In summary, we think our MD experiments have good experimental validation.

18. *The cellular data exploring the role of SERF2 for stress granules are merely phenotyping. How can these be correlated with the in vitro data?
Line 417-419: the authors state that SERF2 may enhance the fluidity of stress granules through LLPS. I do not see why this should be and how this is supported by any of the data.*

Response #18 We thank the reviewer for raising these points. We respectfully want to clarify that our cellular and in vitro results are mechanistically linked. In vivo, we show that SERF2 is critical for stress granule assembly, as SERF2 knockdown impairs granule formation and significantly alters their dynamics, as measured by FRAP in live cell. These phenotypes indicate that SERF2 contributes to the material properties of stress granules similar to what has previously observed for a combined knockdown of core proteins G3BP1 and G3BP2 (Yang et al., 2020, Cell, 181, 325-345. In vitro, we demonstrate that

SERF2 forms condensates with G4 RNAs that are enriched in stress granules. Further, we demonstrate that SERF2 promotes phase separation of G3BP1, a core stress granule protein that highlights SERF2's molecular capacity to modulate LLPS behavior of stress granule components. Together, these results suggest that SERF2 may act as an RNA-scaffold-responsive modulator of stress granule assembly and dynamics.

We have now revised the text (Lines 592-605) to clearly distinguish the evidence supporting the functional connection between SERF2 RNA-mediated condensate formation, and altered fluidity of stress granules. While we agree that further dissection is needed to fully resolve the mechanistic cascade in cells, we consider that our combined data provides good support that SERF2 influences liquid-liquid phase transitions both in vivo and in vitro.

19. *Fig. 7: co-condensation should be shown by using different dyes. It is unclear what the authors mean by SERF2 facilitating phase transitions of G3BP1. What is shown are effects on FRAP recovery. How this relates to fluidity changes and especially how the molecular and structural features discussed before are relevant here is unclear.*

Response #19 To directly address the reviewer's first suggestion, we now include co-condensation data using different dyes Cy-5 labeled G3BP1 and AF488-SERF2, enabling us to directly visualize their colocalization within condensates (Line 585-586; Figure S16a). As shown in Figures 8e and S16a, introduction of SERF2 promotes phase separation of G3BP1 in the presence of RNA, consistent with SERF2 enhancing G3BP1's condensation potential. We have now included the half-life recovery time which shows SERF2 enhances recovery of G3BP1 ~13-fold and ~6-fold in the presence of rG4. Similar facilitative roles have been reported for other stress granule-associated proteins such as CAPRIN-1 and TIA-1 (Yang et al., Cell, 2020, 181, 325-345).

Our FRAP data showing an increased fluorescence recovery rates in the presence of SERF2 or SERF2 and rG4 reflect enhanced internal mobility of G3BP1, indicative of altered material states (i.e., increased fluidity or reduced viscosity). These observations suggest that SERF2 influences the dynamic properties of condensates, a hallmark of phase modulators.

We acknowledge that mechanistically linking molecular interactions to mesoscale condensate properties in tricomponent systems (protein-protein-RNA) is highly complex. Others have noted that such systems are governed by distributed, transient, and multivalent interactions that are challenging to resolve at atomic resolution (Patel et al., Cell, 2015; Banani et al., Nat Rev Mol Cell Biol, 2017; Mittag & Pappu, Curr Opin Cell Biol, 2022). Therefore, while our data support a role for SERF2 in modulating condensate dynamics, detailed structural interpretation of individual interactions in this context remains a future goal requiring further methodological development.

Reviewer #2 (Remarks to the Author)

Summary: This study explores the molecular and structural basis of stress granule formation through high resolution analyses of the small human protein SERF2 and its specific interactions with RNA G quadruplexes (rG4s). The authors employ a combination of NMR spectroscopy, fluorescence assays, single-molecule imaging, and all-atom molecular dynamics (MD) simulations to show how SERF2 engages rG4s and participates in liquid-liquid phase separation. They demonstrate that SERF2's disordered regions mediate multivalent interactions with rG4s, which are important to drive LLPT under physiologically relevant conditions. They further show that SERF2 promotes the formation and fluidity of stress granules in vivo. This work addresses an important and timely question at the interface of structural biology, RNA biophysics, and cellular phase transitions. The authors make a significant stride in this direction.

Response We thank Reviewer 2 for their thoughtful summary of our work and for recognizing its relevance at the interface of structural biology, RNA biophysics, and biomolecular condensates. We appreciate the acknowledgment that our study integrates structural data, functional assays, and computational simulations to reveal the role of SERF2 in G4-RNA recognition and phase separation, both in vitro and in vivo.

However, several key clarifications and additions are needed before the manuscript can be accepted.

Response: We agree that further clarification of a few key mechanistic aspects would enhance the clarity and depth of our manuscript. We have revised the relevant sections and added additional explanations, references, and data wherever appropriate, as detailed in the responses to the individual points below.

Major comments:

1. Why does the system form a hydrated core? This is not addressed completely in the manuscript. The formation of such a ring-like structure seems counterintuitive to having the two components mixed. Is there any experimental evidence to support this structure or is it only seen in simulation?

Response #1 We appreciate the reviewer's insightful question regarding the formation of a hydrated core in our simulation. The ring-like architecture with a solvent-rich center has

Figure (for reviewers' reference): SERF2 (orange); rG4 (blue), PEG (red) and water (grey lines, left panel). Waters are not shown on the right panel for clarity. The ring center measured using ChimeraX on right panel is ~6 nm (empty spheres represent ions).

been observed in our all-atom MD simulations and has not yet been directly observed experimentally for this system. In our MD simulations, G4-RNA molecules tend to cluster into a dense core, driven by SERF2 binding, forming a ring- or shell-like architecture. We have now included a section describing this to address the observation of a ring-like structure (Line 439-447). Though we do not have experimental evidence that this occurs (now included in Line 480-484), we believe the formation of such a solvent-accessible central cavity arises from the molecular geometry and charge complementarity of the system. The positively charged SERF2 molecules dynamically interacting with the polyanionic phosphate backbone and grooves of the G4 structures generating a multivalent interaction network.

SERF2 binds rG4s through electrostatic attraction and groove-specific contacts, but not all rG4 surfaces are simultaneously covered by all SERF2 molecules. Further, not all SERF2 molecules fully interpenetrate the RNA cluster. The partial surface engagement results in a shell-like organization of SERF2 molecules around the RNA-rich domain, which leaves a central region largely unoccupied allowing water molecules to remain, thereby forming a solvent-accessible core. Moreover, additional spatial analysis shows that PEG molecules (see image on the left, PEG molecules are in red) tend to localize at the periphery of the ring, with minimal

presence within the core. This peripheral crowding may further promote water retention in the center, due to excluded volume effects and PEG's poor penetration into dense charged regions.

While direct experimental validation of this architecture is challenging especially given the dynamic and heterogeneous nature of condensates, methods such as PRE-NMR or FRET are, in principle, suited to probing solvent exposure and intra-condensate distances. However, these techniques are significantly limited by the random distribution of spin or fluorescence labels in dense assemblies, orientation dependence, and sample heterogeneity. We consider that our MD simulations offer a powerful means to explore emergent mesoscale organization in biomolecular condensates where experimental resolution is difficult to capture.

2. How do the authors make sure 0.5 us is sufficient time for the simulation? Is this ring-like structure a metastable structure that would disappear after a longer simulation?

Response #2 We thank the reviewer for this important point. We agree that molecular dynamics simulations are always limited by accessible timescales, and it is possible that longer simulations may reveal additional structural transitions. To address the concern regarding metastability, we performed analyses on structural persistence, cluster formation and interaction lifetimes throughout the simulation window. Further, we compared our 0.5 microsecond large-system with three additional small systems running for 1 microsecond and observed a very similar pattern of interactions and ring-shape structures (Line 473-480). The RNA G-quadruplexes remain largely folded throughout these simulations, while SERF2 retain its disordered structure. The SERF2 molecules show sustained association with rG4 clusters over the last 300 ns. Contact maps and cluster stability analyses (Supplementary Figure S11c and S11d) indicate that the configuration reaches a plateau and does not show signs of dissolution, fusion, or

Figure (for reviewers' reference only): We are performing more studies on SERF family proteins and their interaction with different disease related repeat-RNA sequences in a separate project. Here we show the reviewers our unpublished results that demonstrate ring-like structure formation in SERF2 (purple) when it interacts with various repeat RNA (gold) in all-atom and coarse-grained MD simulations. A ring-like structure forms in 1 microsecond for the long-repeat, and an open ring-like structure was obtained for the small-repeat in all-atom MD simulation. Coarse-grained MD of the small repeat at 20 microseconds timescale shows formation of several small ring-like structures.

collapse, suggesting that the observed ring-like structure is kinetically stable within the simulation window.

We note that all-atom simulations of such multicomponent condensates at this scale (containing several RNA and protein molecules plus explicit solvent and crowders) are computationally demanding. To address long-timescale behavior more rigorously, future coarse-grained or hybrid multiscale approaches (e.g., MARTINI + back mapping; see Comment 4) could be useful to explore equilibrium behavior, however limited to intermolecular folded RNA G4s including TERRA rG4s as explained in response to reviewers comment #4 (please see below).

We are currently studying other rG4s using repeat-RNA G4s (hexanucleotide GGGGCC repeat) with time scales 1 to 2 microseconds all-atom simulation to be reported in a future study. Our results showed these ring-like structures are prevalent for short and long-repeat RNA and observed very similar structures and cluster graphs (please see images). Further, we have implemented coarse-grained models to run these systems on a longer timescale i.e. 20 microseconds and have observed that these ring-like structures are stable over these longer timeframes. However, our current edited martini force fields that combine PEG, RNA and protein is not yet validated and need further improvisation and require direct comparison with all-atom MD results.

3. What is the role of PEG? Are there any PEG-protein/RNA interactions in the system? Is all the PEG outside the droplet or is it present within the hydrophilic core?

Response #3 We thank the reviewer for raising this important question. Yes, we do observe PEG interaction with protein and G4 molecules in the condensed structures, and discussion of this point is now included in the revised manuscript (Line 448-456). Visual inspection of simulation trajectories, along with contact frequency analysis (see revised Figure 4d and 4e), revealed that PEG molecules do indeed interact with both SERF2 and rG4 molecules in the condensed phase. We have now included a graph in Figure 4e to show the distribution of PEG within 6 Å of protein/RNA and some snapshots are shown below for the reviewers' reference. We also observed that PEG molecules are excluded from the solvent-rich core region of the ring-like condensate. Instead, they tend to localize

Figure (for reviewers reference): SERF2 (orange); rG4 (blue), PEG (red) and water (grey lines). In the condensed phase (left panel) several protein and RNA molecules are in contact with PEG. In the dilute structures (right panel), while some are in contact with PEG others are not.

near the outer protein-RNA shell. This PEG partitioning behavior is likely driven by steric hindrance and limited permeability through the dense protein-RNA network at the periphery. Thus, our data support a dual role for PEG in this system. An indirect promotion of LLPT through crowding effects, enhancing the multivalent interactions of SERF2 with rG4 RNA and limited, yet notable, interactions with biomolecules, especially in the outer regions of the condensate, which may stabilize the overall assembly.

4. Why do the authors not use a backmapping strategy to ensure that the system is pre-equilibrated before starting simulations at atomistic resolution? Simulating the system using MARTINI could allow for the system to equilibrate faster which could then be used to backmap and study atomic interactions.

Response #4 We sincerely thank the reviewer for the thoughtful suggestion regarding the use of coarse-grained (CG) modeling and backmapping strategies to accelerate system equilibration prior to atomistic simulations. We agree that CG approaches such as those enabled by the MARTINI force field are powerful for capturing long-timescale assembly behaviors. However, to our knowledge, the MARTINI 3 force field is not yet validated to support simultaneous simulation of protein-RNA-PEG ternary systems. While the MARTINI 3 force field includes updated parameters for proteins and nucleic acids (Souza et al., Nat. Methods, 2021), it does not currently provide a reliable, unified framework for structured RNAs such as G-quadruplexes. We have confirmed this limitation in consultation with MARTINI developers and through user forums such as martini-consultation GitHub threads. Further, in coarse-grained model, we encounter issues with G4 stability in particular for those molecules that are not intramolecular, and unfortunately for our studied system TERRA rG4 experimental structures are intermolecular and available either in dimeric and tetrameric states. We have tested our CG force field on a dimeric G4 experimental structure (PDB ID: 8X0S), and observe its disintegration in CG MD simulation (please see image below). This restricts us to apply CG-MD simulation for a system comprising SERF2, TERRA rG4 and PEG crowders.

Nevertheless, we have taken steps in this direction applying the MARTINI 3 parameters to model a simplified intramolecular RNA system (GGGGCC repeat RNA), along with PEG and a disordered protein. In these test simulations, we did observe ring-like structures reminiscent of those seen in our all-atom models. However, a direct correspondence between CG and atomistic results could not be reliably established, likely due to the known over-sticky behavior of MARTINI interactions, especially for nucleic acids and disordered protein regions (Best et al., JCTC, 2022; Takada et al., Biophys. Rev., 2021).

We are currently validating our results through systematic comparison with experimental data and all-atom controls, which is ongoing. As a preliminary insight for the reviewer, we have now included a comparative structural snapshot and quantitative analysis (unpublished results, please see image in response#2) showing the ring-like architectures from both CG-MD and all-atom MD for the test system. We aim to fully develop and deploy this CG-to-atomistic workflow in future iterations, once a rigorously validated force field becomes available for ternary systems that include PEG, RNA, and protein.

5. How dynamic is the ring-like structure? Do SERF/RNA molecules within the structure exchange with the surroundings or not? If not, how do the authors conclude that these are liquid-like assemblies?

Response #5 We thank the reviewer for raising this point. We analyzed the dynamics and exchange behavior of SERF2 and rG4 RNA molecules within the ring-like condensates observed in our all-atom MD simulations.

To assess molecular mobility within the condensed phase, we performed mean squared displacement analysis of SERF2 molecules in the simulated condensates. The MSD plots show linear behavior with average diffusion constant of $1.07 \mu\text{m}^2/\text{s}$, indicating that the molecular motion is not arrested (which would correspond to $\alpha \sim 0$), but rather partially diffusive and dynamic, consistent with the viscous liquid-like nature of these assemblies (data included in revised Figure 4g, Line 504-509).

Additionally, visual inspection of simulation trajectories revealed reorganization and exchange of protein components across the ring perimeter. These behaviors are consistent with reversible, multivalent interactions between disordered SERF2 regions and rG4s, rather than irreversible aggregation or gel-like solidification. We have included some discussion of these points in the revised manuscript.

6. The authors observe striking ring-like assemblies in MD simulations involving multiple SERF2 and rG4 molecules. These structures are interpreted as representative of early phase condensates. Are there any microscopy data that might support this structural formation?

Response #6 The ring-like assemblies we observe in our all-atom MD simulations are approximately ~ 6 nm in diameter (Figure S11a), which places them well below the resolution limit of conventional light or fluorescence microscopy. Due to their small size and the dynamic, densely packed nature of the condensates, it is currently not feasible to directly validate these structures experimentally using these or other standard imaging approaches.

We also considered biophysical techniques such as FRET and paramagnetic relaxation enhancement (PRE) NMR to obtain experimental validation of our ringlike structures, but these methods also show limitations. Specifically, the high local concentration of SERF2 and RNA molecules within the condensates creates a dense and heterogeneous environment that complicates accurate FRET distance measurements or clean interpretation of PRE data, both of which rely on relatively sparse and defined interactions. Therefore, we interpret these ring-like structures as atomistic models representing early, nascent interactions that may underlie the formation of larger, dynamic condensates observed experimentally. These structural insights are valuable as they help bridge the gap between molecular determinants and mesoscale phase behavior, even though direct experimental resolution at the nanometer scale remains out of reach at present. We have clarified these points in the revised to temper the interpretation of our MD structures and highlight the technical challenges in validating them experimentally. As noted in the response to Query-2, we observe similar ringlike structure in both all atom and coarse-grained MD simulations with other quadruplexes.

7. The MSD vs. lag time plots for the atomistic condensed phase should be shown. Do these plots show an exponent of 1 indicative of diffusive motion or is the motion of SERF arrested in their simulations?

Response #7 In response to this comment, we have now computed MSD vs. lag time plot. The MSD plots exhibit a linear trend (average MSD $\sim 6.4 \mu\text{m}^2/\text{s}$), consistent with sub-

diffusive motion, a feature of molecular dynamics in dense, multivalent condensates. This behavior indicates that SERF2 molecules are not arrested, but rather experience restricted mobility, likely due to transient multivalent interactions with rG4 molecules within the condensate.

We now include these MSD plots and fitted exponents as a new panel in Figure 4g, and we describe this analysis in the Results section (Lines 504-509) and in the revised methods section. These data support the conclusion that the condensates retain liquid-like behavior, albeit with a viscoelastic and crowded internal environment, an expected property of protein-RNA phase-separated structures, as described in: Harmon et al., *Nature Structural & Molecular Biology*, 2017; Patel et al., *Cell*, 2015; Wei et al., *Cell*, 2017.

8. *Within the condensed phase, do the RNA chains maintain their quadruplex structure? In the dilute phase, the authors observe less stability than experiments. Does the condensed phase stabilize the structure more or favor unfolding?*

Response #8 To address these questions, we performed additional analysis of rG4 stability. We calculated the Hoogsteen hydrogen bonds as a function of simulation time. The results now included in Figure 4f show an average ~40 Hoogsteen bonds for the 30 RNA molecules in the condensed structure suggesting the integrity of rG4 structures were retained in our simulation. We were unable to directly compare these results with experimental data as CD absorption is sensitive to sample turbidity.

These findings are consistent with prior studies suggesting that macromolecular crowding can stabilize G-quadruplex structures (Gao et al., *ACS Omega*. 2023, 8:14342-14348). We now include a summary of these analyses in Figure 4f and updated the main text accordingly (Lines 485-496).

Minor comments:

The section regarding whether crowders are used in the MD simulation in the condensed phase is unclear. Going through the methods, it is clear that crowders were added, but the text in the results section is very confusing. The authors could simply state that they add the crowder to match the NMR conditions.

Response# We appreciate the reviewer's careful reading and agree that the description of crowder usage in the condensed-phase MD simulations could be made clearer. To clarify: we added PEG crowders explicitly in our atomistic simulations of the condensed phase to match the experimental conditions used for in vitro phase separation assays. We have now revised the Results section (Lines 448-456) to clearly state the rationale and method for incorporating PEG crowders and have also added a clarification in the Methods section. In addition, we have clearly stated in the results sections where crowders were added.

Reviewer #3 (Remarks to the Author):

This study examined the structural details of the protein SERF2 and its interactions with RNA G-quadruplexes, in the context of phase separation. The authors first showed that SERF2 can specifically bind to rG4 structures. Using high-throughput sequencing methods, they showed that the SERF2 prefer to interact G-rich RNAs that are likely to form stable rG4 structures, including several known rG4 sequences. In vitro, SERF2 can form liquid droplets with G4 or HeLa cell total RNAs. However, SERF2-rG4 droplets are less dynamic, both protein and RNA exchange slower buffer compared to SERF2-total RNA droplets.

Using NMR chemical shifts and NOE restraints, the authors defined the structure of SERF2 in solution, which consists of a helical region flared by disordered N- and C-term regions. Protein

CSP analysis and relaxation measurements revealed that both the N- and C-term disordered regions mediate SERF2's interaction with the model rG4 TERRA, where the N-terminus drives the binding, and the C-terminus coordinate the complex formation. Size-exclusion chromatography showed that rG4 can form 1:1 and 1:2 multimeric complexes with SERF2. Using docking program, the authors proposed model structures for 1:1 and 1:2 TERRA-SERF2 complex and subsequently refined them using all-atom simulation. Consistent with the NMR results, the complex structure is mediated by the N-term disordered residues in SERF2. The complex structures revealed details such as H-bonds between R and K residues in SERF2 and bottom guanines in TERRA, additional amino acids are involved in binding in the 1:2 complex, and that protein-RNA bonding can disrupt G-G pairs to destabilize rG4 structure. The destabilization of rG4 structure upon binding to protein is further verified by CD, though the extent was overestimated in the MD simulation. To verify amino acids involved in SERF2-rG4 binding, the authors carried out mutational study and confirmed the importance of these key residues identified in NMR and MD studies.

Next, to study the RNA-protein co-phase separation, the authors simulated RNA-protein condensate using all-atom simulation. In addition to unbound rG4 and SERF2-TERRA rG4 oligomers, a ring-link higher order structure was identified. FRET data suggested that SERF2 adopts a more compact structure within the condensate, consistent with MD simulation results.

In multiple cell lines, the authors showed that depletion of SERF2 drastically reduce the number of induced stress granules. The extent of impact is comparable to knockdown of G3BP1+G3BP2. Additionally, cellular granules formed under SERF2 depleted conditions are more like aggregates than condensates.

We thank the reviewer for the careful reading and thoughtful comments. We appreciate the positive assessment of our integrative approach combining structural, biophysical, and cellular studies to elucidate SERF2-rG4 interactions. Below we address each point in detail.

Comments:

1. Overall, I think this manuscript presents a well-rounded study of the SERF2-rG4 interactions in both solution and phase separation. The experimental and simulation studies were designed to complement each other and combine to produce an in-detail structural picture of the complex in various conditions. I found the interpretation of data to be reasonable. However, I do have a few concerns, see below:

The authors need to specify the TERRA RNA sequence used in each assay and provide justification for not using the same TERRA sequence. I.e., in the 1D ¹H NMR titration, the authors apparently used TERRA12, whereas in the HADDOCK/All-atom modeling, the authors switched to a shorter TERRA10 sequence. This is problematic because various TERRA sequences can form different structures: TERRA12 forms a dimeric, 3-layer G4 structure while TERRA10 forms a tetrameric, 5-layer structure. Because of the distinct structures, the SERF2-rG4 binding modes might not be the same. Had the author used the same sequence in these studies, we would have been able to compare amino acid/RNA bases that participate in the binding to see if they are consistent.

Response #1 We appreciate the reviewer's insightful observation. We have now clearly stated the TERRA sequences used in each assay both within the Methods and Results sections. TERRA12 (5'-UUAGGGUUAGGG-3') was used in NMR titration experiments due to its isolated and well-resolved NMR signals for all the guanine imino protons (Martadinata and Phan, 2009, J Am Chem Soc 131, 2570-2578), which allowed imino proton mapping of chemical shift perturbations with SERF2 titration. We have tested different length TERRA rG4 that include TERRA10, 12, 23 and 60 to compare their binding mode and affinity for SERF2. As commented by the reviewer, TERRA12 (PDB: 2KBP) and TERRA10 (PDB: 2M18) respectively form dimeric and tetrameric structures, but we found

that both show a similar binding interface on SERF2 (Figure S5). Further, we found G4, G5, G9 and G10 in both these RNA molecules show chemical shift perturbations upon binding to SERF2. Our simulation results show TERRA10 is relatively stable as compared to TERRA12 in the chosen force field for simulation (additional results are included in Figure S10a), and therefore we decided to use TERRA10 for our simulation but for the reasons stated above consider that many if the findings we obtained for TERRA10 should also apply to TERRA112. We further included supplementary figures (Figures S5 and S6) comparing the binding of different TERRA molecules that include 10, 12, and 60 nucleotides to SERF2 and observed a similar binding interface at low RNA concentrations.

2. *I wonder if the ring-like tertiary structure seen in the condensed phase simulation is a result of the specific protein and RNA numbers and their ratio in this set up, or a general phenomenon. The authors should attempt the same experiment with different protein/RNA counts to test this.*

Response #2 This is an excellent point. We have now tested the possible formation of a ring-like structure by setting up two additional microsecond long MD systems using different stoichiometries, 1:2 and 2:1 SERF2:rG4 in addition to the 1:1 stoichiometry presented in the original submission. These data are now included in the revised manuscript (Lines: 473-480) as Supplementary Figure S12. The data shows the formation of a ring-like structure in 1:2 protein:rG4 system, and a partial ring-like structure was observed for the 2:1 system (Figure S12). Together, the new results suggest, the ring-structure persists at different protein-rG4 stoichiometries.

3. **Minor comments:**

Ca, Cb secondary shifts should be reported from the SERF2 assignments, in order to provide further evidence for secondary structures in the protein.

Response #3 We have submitted the Ca and Cb chemical shifts to BMRB database and the SERF2 PDB co-ordinates are submitted to PDB databank under 9DT0. We have reported secondary structure using TALOS predicted dihedrals from Ca and Cb chemical shifts in our NMR assignment article *Biomol NMR Assign.*, 2024, 18, 51-57.

4. *PEG molecules are shown in figure 4a only at 0 μ s but not 0.5 μ s. It would be interesting to see how the PEG molecule exists in the simulated condensed phase and whether they interact with the protein.*

Response #4 We thank the reviewer for this comment. We have now included additional figures (Figures 4d and S11a) to illustrate the spatial distribution of PEG molecules as a function of time (0 μ s and 0.5 μ s). The analyzed results are included in Lines: 448-456.

5. *In Figure 4c the two 2:1 complex structure in the simulated condensed phase and the dilute phase structures are oriented in a way that is difficult to see. Slight adjustment should help to draw a better comparison between them.*

Response #5 We have adjusted the orientation and zoom of the molecular models in Figure 4c to enhance visibility of the RNA-protein interface and overall architecture.

6. *The data in figure 7e does not appear to support the claim that SERF2 enhances G3BP1 fluidity. While in the microscopy image the droplet with SERF2 does appear to recover faster, statistical comparison in 7e suggest that the difference is within error.*

Response #6 We regret for not clarifying the statistical differences in G3BP1 and SERF2 recovery rates in the absence or presence of RNA. We have included the half-time recovery, and show that SERF2 influences ~13-fold faster recovery for G3BP1. In the

presence of rG4, the G3BP1 recovery was ~6 fold faster, whereas the recovery is slightly faster (~1.3 fold) in the presence of total RNA (revised figure 8e). We agree with reviewer that not much difference that is statistically significant was observed in SERF2 recovery rates under these various conditions indicating SERF2 remain in a dynamic state in all conditions. We have clarified this in the results to make it clear that SERF2 or SERF2 and rG4 together enhance G3BP1 fluidity and revised text are included in results (Lines: 592-605).

7. *“condensation is mainly due to an increase in the valency of interactions driven by crowding” – I think the term valency refers to the potential number of sites for interaction. So, crowding does not change that (it does not change the (primary) structure). So, I would rephrase this.*

Response #7 We thank the reviewer for the correction. The revised sentence now reads: “Condensation is facilitated by an increase in the effective interaction network density under molecular crowding, which enhances multivalent interactions without altering the intrinsic valency of individual molecules (Lines: 428-430).”

Reviewer #4 (Remarks to the Author):

Reviewer #5 (Remarks to the Author):

Response to Reviewer #4 and Reviewer #5:

We thank Reviewer #4 and Reviewer #5 for their contributions. We appreciate their time and efforts in evaluating our manuscript and contributing to the peer review process.

Response to reviewers' comments

We thank the reviewers for their positive assessment of our revised manuscript and for their constructive suggestions that have helped us further improve the clarity of our presentation. Below, we address the remaining minor points raised by Reviewer #3.

In Figure S5, some of the green bars for SERF2–TERRA CSPs are covered by the blue bars; re-plotting so both are visible would be helpful.

Response: In the revised Figure S5, we have replotted the data side-by-side to ensure that both the bars for SERF2-TERRA chemical shift perturbations are clearly visible.

The new Figure S6 would benefit from including also a set of peak intensity plots, which would help to show if the same residues are involved in binding TERRA RNAs of different lengths.

Response: We have added a new set of peak intensity plots to Figure S6 to help visualize how different residues are involved in binding TERRA RNAs of different lengths (10 versus 60 nucleotides), as requested. We exclude plotting the peak intensity ratios for other spectra shown in Figure S6 that include TERRA23 and TERRA60 at high RNA concentration, because most of the ¹H-¹⁵N peaks in these experiments are broadened with a low signal-to-noise ratio, making reliable peak height integration difficult. We believe this addition improves the interpretability of the results and addresses the reviewer's suggestion.

We appreciate Reviewer #3's attention to presentation details, which has helped us to enhance the clarity and completeness of our figures.